# Multi-Objective Coverage Bayesian Optimization
## (`MOCOBO`)

**Natalie Maus**[*]    **Kyurae Kim**    **Yimeng Zeng**    **Haydn Thomas Jones**    **Fangping Wan**
**Marcelo Der Torossian Torres**    **Cesar de la Fuente-Nunez**    **Jacob R. Gardner**

University of Pennsylvania
Philadelphia, PA, USA

## Abstract

In multi-objective black-box optimization, the goal is typically to find solutions that optimize a set of $T$ black-box objective functions, $f_1, \ldots f_T$, simultaneously. Traditional approaches often seek a single Pareto-optimal set that balances trade-offs among all objectives. In contrast, we consider a problem setting that departs from this paradigm: finding a small set of $K < T$ solutions, that collectively "cover" the $T$ objectives. A set of solutions is defined as "covering" if, for each objective $f_1, \ldots f_T$, there is at least one good solution. A motivating example for this problem setting occurs in drug design. For example, we may have $T$ pathogens and aim to identify a set of $K < T$ antibiotics such that at least one antibiotic can be used to treat each pathogen. This problem, known as coverage optimization, has yet to be tackled with the Bayesian optimization (BO) framework. To fill this void, we develop Multi-Objective Coverage Bayesian Optimization (`MOCOBO`), a BO algorithm for solving coverage optimization. Our approach is based on a new acquisition function reminiscent of expected improvement in the vanilla BO setup. We demonstrate the performance of our method on high-dimensional black-box optimization tasks, including applications in peptide and molecular design. Results show that the coverage of the $K < T$ solutions found by `MOCOBO` matches or nearly matches the coverage of $T$ solutions obtained by optimizing each objective individually. Furthermore, in *in vitro* experiments, the peptides found by `MOCOBO` exhibited high potency against drug-resistant pathogens, further demonstrating the potential of `MOCOBO` for drug discovery. All of our code is publicly available at the following link: `https://github.com/nataliemaus/mocobo`.

## 1 Introduction

Bayesian optimization (BO; [1–3]) is a general framework for sample-efficient optimization of black-box functions. By using a probabilistic surrogate model, such as a Gaussian process (GP; [4]), Bayesian optimization balances exploration and exploitation to identify high-performing solutions with a limited number of function evaluations. BO has been successfully applied in a wide range of domains, including hyperparameter tuning [5, 6], A/B testing [7], chemical engineering [8], drug discovery [9], and more.

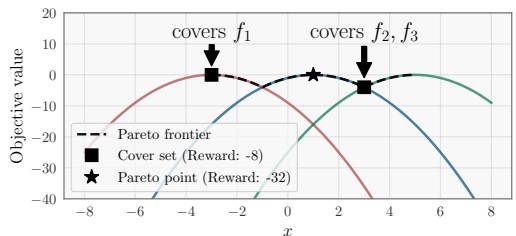

Figure 1: Traditional multi-objective optimization for $T$ objectives might select any point along the Pareto frontier, but in some situations like this any Pareto optimal point performs poorly on at least one objective. In situations where multiple $K < T$ solutions are allowed (■), we can sometimes optimize all objectives well. Note that this is a simplified schematic meant to illustrate intuition.

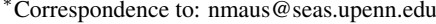

[*]Correspondence to: nmaus@seas.upenn.edu

39th Conference on Neural Information Processing Systems (NeurIPS 2025).

For application domains such as drug discovery, however, the setup assumed by vanilla BO, where there is only one clear objective, is often insufficient. Consider the example of designing antibodies for treating a broad spectrum of pathogens. Here, the activity against each individual pathogen serves as an objective, each equally important. Ideally, one would seek to optimize all objectives simultaneously by obtaining a single broad-spectrum antibiotic. When this is not possible, however, we should seek antibiotics that are potent against a large enough group of pathogens. By identifying a set of such solutions—a *covering set*—we would obtain a set of drugs that are *together* lethal against the set of pathogens.

At first, this setup may be reminiscent of multi-objective optimization, which has been tackled under the black-box setup via multi-objective BO (MOBO; [10–16]). These methods search for Pareto-optimal solutions that balance the trade-offs between the individual objectives, enabling practitioners to *post hoc* select a *single* solution lying on the Pareto front based on their preferences. In cases where some objectives are completely incompatible with each other, however, selecting a single solution on the Pareto front cannot simultaneously satisfy performance requirements across those objectives. This is illustrated in Figure 1, where no individual solution can provide acceptable performance across all objectives due to the extreme trade-offs between objectives.

Instead, our problem is distinctly described as the *coverage optimization* problem. More formally, consider a set of $T$ distinct objectives. Our goal is to obtain a set-valued solution of size $K < T$, where the set "covers" the $T$ objectives: For each objective, this $K$-element covering set should contain at least one solution that performs similarly to the maximizer of that objective. In the context of drug design, this translates to identifying $K$ drugs such that each of the $T$ pathogens is effectively addressed by at least one drug in the set. This setting is particularly relevant to applications where the size of $K$ is associated with downstream cost. For instance, in drug discovery, the number of compounds that can be synthesized is fundamentally constrained by the capacity of the facility.

Previously, the coverage optimization problem has been tackled in [17–19] through gradient-based algorithms, while for the black-box setup, Liu et al. [20] proposed an evolutionary algorithm. Over the years, variants of BO have demonstrated superior performance over evolutionary algorithms in high-dimensional [15, 21, 22] and multi-objective problems [14], which has been instrumental for its success in drug discovery [22–25]. Therefore, it is natural to ask how we could harness the strengths and flexibility of BO to attack the coverage optimization problem.

In this work, we develop Multi-Objective Coverage Bayesian Optimization (MOCOBO)—a BO-based method for solving the coverage optimization problem with high-dimensional black-box objectives. MOCOBO primarily consists of a coverage-optimization analog of the celebrated expected improvement (EI; [26]) acquisition function, where the covering set is greedily constructed via submodular optimization [27]. We evaluate the resulting algorithm on realistic optimization tasks: drug discovery over molecules and peptides, and tuning of image processing pipelines. Baselines include the method of [20] and MOBO.

Our contributions are summarized as follows:

- We propose MOCOBO, a coverage optimization algorithm for black-box objectives that can deal with structured, high-dimensional search spaces, and scale to large numbers of function evaluations.
- We experimentally validate MOCOBO on challenging, high-dimensional optimization tasks including structured drug discovery over molecules and peptides, demonstrating its ability to consistently outperform state-of-the-art BO methods in identifying high-performing covering sets of solutions.
- We demonstrate empirically that the small sets of $K < T$ solutions found by MOCOBO consistently nearly match the performance of a larger set of $T$ solutions individually optimized for each task.
- We demonstrate the potential of MOCOBO for drug discovery using an *in vitro* experiment showing it produces potent antimicrobial peptides that cover 9 out of 11 drug resistant or otherwise challenging to kill pathogens, with moderate activity on 1 out of 11.

## 2  Background

**Bayesian Optimization.**   To solve the black-box optimization problem, BO operates by iteratively selecting a query according to a certain search policy [3] and observing the objective value on the query. The observations that have been obtained up to iteration $n \geq 0$ are collected into a dataset $D_n$ to form a surrogate model, typically GP, which supplies information to the policy. Most policies are

chosen to maximize a function known as *acquisition function* [3] such that, at each step $n \geq 0$, the query $\mathbf{x}_{n+1}$ is found as

$$\mathbf{x}_{n+1} = \arg\max_{\mathbf{x} \in \mathcal{X}} \alpha\left(\mathbf{x}; D_n\right) , \tag{1}$$

where EI [26] is a popular example of an acquisition function. In our case, the optimization problem is over covering sets. Naturally, an acquisition function for this setting is needed.

**Trust Region Bayesian Optimization (TuRBO).** For high-dimensional problems, the basic scheme of solving Equation (1) is no longer effective. Local BO methods [21] address this issue by restricting the search space to a trust region that is adjusted adaptively. This simple modification has been shown to be effective and has been employed in various setups. In high-dimensional MOBO problems specifically, a variant of TuRBO [21] referred to as TuRBO-$M$ has been shown to be effective [14, 28]. TuRBO-$M$ works by running $M$ parallel local optimization runs, each maintaining its own dataset $D_i$ and surrogate model. Each local optimizer proposes candidates within a hyper-rectangular trust region $\mathcal{T}_i$, which is taken to be a rectangular subset of the space $\mathcal{X}$ centered on the incumbent $\mathbf{x}_i^+$. The side of each $\mathcal{T}_i$ is chosen to have a length $\ell_i \in [\ell_{\min}, \ell_{\max}]$. If an optimizer improves the incumbent for $\rho_{\text{succ}}$ consecutive iterations, $\ell_i$ expands to $\min(2\ell_i, \ell_{\max})$. If not improved in $\rho_{\text{fail}}$ iterations, $\ell_i$ is halved. Optimizers are restarted if $\ell_i$ drops below $\ell_{\min}$. We will later incorporate this scheme for solving the coverage optimization problem.

# 3 Multi-Objective Coverage Bayesian Optimization

## 3.1 Coverage Optimization

Consider $T > 1$ objectives $f_1, \ldots, f_T$, each defined as $f_t : \mathcal{X} \to \mathbb{R}$ for each $t = 1, \ldots, T$, all sharing the same input domain $\mathcal{X}$. For some user-defined parameter $K \in \{1, \ldots, T-1\}$, we consider the task of finding a set of $K$ solutions $S^* = \{\mathbf{x}_1^*, \ldots, \mathbf{x}_K^*\}$ that "covers" the $T$ objectives $\{f_t\}_{t=1,\ldots,T}$. Specifically, the set $S^* \subseteq \mathcal{X}$ "covers" the $T$ objectives if, for each objective $f_t \in \{f_1, \ldots, f_T\}$, there is at least one solution $\mathbf{x}_k^* \in S^*$ for which $f_t$ is well optimized by $\mathbf{x}_k^*$. We evaluate how well a set of $K$ points $\{\mathbf{x}_1, \ldots, \mathbf{x}_K\}$ "covers" the $T$ objectives using the following "coverage score" Formally,

$$c(\{\mathbf{x}_1, \ldots, \mathbf{x}_K\}) = \sum_{t=1}^{T} \max_{k=1}^{K} f_t(\mathbf{x}_k). \tag{2}$$

Formally, we seek a set $S^* := \{\mathbf{x}_1^*, \ldots, \mathbf{x}_K^*\}$ such that:

$$S^* = \arg\max_{\{\mathbf{x}_1, \ldots, \mathbf{x}_K\} \subseteq \mathcal{X}} c(\{\mathbf{x}_1, \ldots, \mathbf{x}_K\}). \tag{3}$$

The coverage score in (2) only credits a single solution for each objective. If, for example, $f_t$ is maximized by one of the $\mathbf{x}_i$ in the solution set, improving the value of $f_t$ on some other $\mathbf{x}_{j \neq i}$ becomes irrelevant so long as $f_t(\mathbf{x}_i) \geq f_t(\mathbf{x}_j)$. In the setting where $K = 1$, the coverage score collapses into a trivial linearization of the objectives, and true multi-objective BO methods should be preferred. In the setting where $K = T$, the coverage score is trivially optimized by maximizing each objective independently. This problem has previously been formulated in [17–20], where we will proceed by developing a new BO-based solution.

## 3.2 Multi-Objective Coverage Bayesian Optimization (MOCOBO)

In this section, we propose MOCOBO - an algorithm which extends Bayesian optimization to the problem setting above. On each step of optimization $s$, we use our current set of all data evaluated so far $D_s = \{(\mathbf{x}_1, \mathbf{y}_1), \ldots, (\mathbf{x}_n, \mathbf{y}_n)\}$ to define the best covering set $S_{D_s}^* = \{\mathbf{x}_1^{*(s)}, \ldots, \mathbf{x}_K^{*(s)} \in D_s\}$ found so far. Here $\mathbf{y}_i = (f_1(\mathbf{x}_i), \ldots, f_T(\mathbf{x}_i))$ and $n$ is the number of data points evaluated so far at step $s$. Following Equation (3), we define $S_{D_s}^*$ as follows:

$$S_{D_s}^* = \arg\max_{\{\mathbf{x}_1, \ldots, \mathbf{x}_K\} \subseteq D_s} c(\{\mathbf{x}_1, \ldots, \mathbf{x}_K\}) = \arg\max_{\{\mathbf{x}_1, \ldots, \mathbf{x}_K\} \subseteq D_s} \sum_{t=1}^{T} \max_{k=1}^{K} f_t(\mathbf{x}_k). \tag{4}$$

Here, $c(S_{D_s}^*)$ denotes the best coverage score found by the optimizer after optimization step $s$.

### 3.2.1 Candidate Selection with Expected Coverage Improvement (ECI)

Given $S^*_{D_s}$, our surrogate model's predictive posterior $p(y \mid \mathbf{x}, D)$ induces a posterior belief about the improvement in coverage score achievable by choosing to evaluate at $\mathbf{x}$ next. We naturally extend the typical expected improvement (EI; [26]) acquisition function to the coverage optimization setting by defining expected coverage improvement (ECI):

$$\text{ECI}(\mathbf{x}) = \mathbb{E}_{p(y\mid\mathbf{x},D)}[\max(0, c(S^*_{D_s\cup\{(\mathbf{x},\mathbf{y})\}}) - c(S^*_{D_s}))]. \tag{5}$$

Here $c(S^*_{D_s})$ is the coverage score of the best possible covering set from among all data observed $D_s$ – as we shall see, constructing this set $S^*_{D_s}$ will be our primary challenge. $c(S^*_{D_s\cup\{(\mathbf{x},\mathbf{y})\}})$ is the coverage score of the best possible covering set after adding the observation $(\mathbf{x}, \mathbf{y})$. Thus, ECI gives the expected improvement in the coverage score after making an observation at point $\mathbf{x}$. We aim to select points during acquisition that maximize ECI. Note that the name ECI is shared with a method proposed by Malkomes et al. [29] for multi-objective experimental design, but their method targets an entirely different notion of "coverage." (See Section 5 for more details.)

We estimate ECI using a Monte Carlo (MC) approximation. To select a single candidate $\hat{\mathbf{x}}$, we sample $m$ points $P = \{\mathbf{p}_1, \mathbf{p}_2, ..., \mathbf{p}_m\}$. For each sampled point $\mathbf{p}_j$, we sample a realization $\hat{\mathbf{y}}_j = (\hat{f}_1(\mathbf{p}_j), \ldots, \hat{f}_T(\mathbf{p}_j))$ from the GP surrogate model posterior. We leverage these samples to compute an MC approximation to the ECI of each $\mathbf{p}_j$:

$$\text{CI}(\mathbf{p}_j) = \max(0, c(S^*_{D_s\cup\{(\mathbf{p}_j,\hat{\mathbf{y}}_j)\}} - c(S^*_{D_s}))). \tag{6}$$

Here, $c(S^*_{D_s\cup\{(\mathbf{p}_j,\hat{\mathbf{y}}_j)\}})$ is the approximation of the coverage score of the new best covering set if we choose to evaluate candidate $\mathbf{p}_j$, assuming the candidate point will have the sampled objective values $\hat{\mathbf{y}}_j$. We select and evaluate the candidate $\mathbf{p}_j$ with the largest expected coverage improvement.

### 3.2.2 Greedy Approximation of Best Observed Covering Set

The candidate acquisition method in Section 3.2.1 utilizes the best covering set of $K$ points among all data collected so far, $S^*_{D_s}$. On each step of optimization $s$, MOCOBO must therefore construct $S^*_{D_s}$ from all observed data $D_s$, as $S^*_{D_s}$ may change on each step of optimization after new data is added to $D_s$.

**Lemma 3.1** (NP-hardness of Optimal Covering Set). *Let $T, K$ be finite positive integers such that $K < T$. Let $f_1, \ldots, f_T$ be real valued functions. Let $D_s = \{(\mathbf{x}_1, \mathbf{y}_1), \ldots, (\mathbf{x}_n, \mathbf{y}_n)\}$ be a dataset of $n$ real valued data points such that for all $\mathbf{x}_i$, $\mathbf{y}_i = (f_1(\mathbf{x}_i), \ldots, f_T(\mathbf{x}_i))$. Let $S^*_{D_s}$ be the optimal covering set of size $K$ in $D_s$ as defined in Equation (4). Then, constructing $S^*_{D_s}$ is NP-hard.*

*Proof.* The proof is in Appendix I. □

Since constructing $S^*_{D_s}$ is NP-hard, we use an approximate construction of $S^*_{D_s}$ on each step of optimization $s$. We present Algorithm 1, which is based on greedy submodular optimization [27], that provides a $(1-\frac{1}{e})$-approximation of $S^*_{D_s}$ after an execution time of $O(nKT)$.

**Theorem 3.2.** *The set $A^*_{D_s}$ output by Algorithm 1 satisfies $c(A^*_{D_s}) \geq (1 - \frac{1}{e}) c(S^*_{D_s})$.*

*Proof.* The proof is in Appendix J. □

---

**Algorithm 1:** Greedy $(1 - \frac{1}{e})$-Approximation for Finding $S^*_{D_s}$ (Incremental Strategy)

**Require:** Dataset $D_s$, observed values $f_t(\mathbf{x})$ for all $t \in \{1, \ldots, T\}$ and $\mathbf{x} \in D_s$, set size $K$.
  Initialize $A \leftarrow \emptyset$. {Start with an empty set.}
  Compute the initial coverage score for $A$: $c(A) \leftarrow 0$.
  **for** $k = 1$ to $K$ **do**
    Initialize $\mathbf{x}_{\text{best}} \leftarrow$ None and $\Delta_{\text{best}} \leftarrow -\infty$.
    **for** $\mathbf{x} \in D_s \setminus A$ **do**
      Compute marginal coverage of adding $\mathbf{x}$ to $A$:
$$\Delta c \leftarrow \sum_{t=1}^{T} \max\left(\max_{\mathbf{x}'\in A} f_t(\mathbf{x}'), f_t(\mathbf{x})\right) - c(A).$$
      **if** $\Delta c > \Delta_{\text{best}}$ **then**
        $\mathbf{x}_{\text{best}} \leftarrow \mathbf{x}$ and $\Delta_{\text{best}} \leftarrow \Delta c$.
      **end if**
    **end for**
    Update $A \leftarrow A \cup \{\mathbf{x}_{\text{best}}\}$, $c(A) \leftarrow c(A) + \Delta_{\text{best}}$.
  **end for**
  **Output:** $A^*_{D_s} \leftarrow A$.

---

**Time Complexity of Algorithm 1.** The algorithm iterates $K$ times to construct the covering set. In each iteration, it evaluates at most $n$ candidate points, and for each candidate, it computes the incremental coverage score by iterating over $T$ objectives. The execution time complexity is thus $O(K \cdot n \cdot T)$. For practical applications where we can assume relatively small $K$ and $T$, the execution time is approximately $O(n)$. For an empirical evaluation of the execution time, see Appendix B.3.

**Corollary 3.3** (Algorithm 1 is the best possible approximation of $S^*_{D_s}$)**.** *There is no polynomial execution time algorithm that provides a better approximation ratio unless $P = NP$.*

*Proof.* This follows from the fact that we reduced from Max $k$-cover, for which a better approximation ratio is not practically achievable unless $P = NP$ [30]. □

### 3.2.3 Extending ECI to the Batch Acquisition Setting (q-ECI)

In batch acquisition, we select a batch of $q > 1$ candidates for evaluation. Following recent work on batch EI [31, 32], we define q-ECI, a natural extension of ECI to the batch setting:

$$q\text{-ECI}(\mathbf{X}) = \mathbb{E}_{p(\mathbf{Y}|\mathbf{X},D)}\Big[\max\Big\{0, \max_{r=1,\ldots,q} c\big(S^*_{D_s \cup \{(\mathbf{x}_r, \mathbf{y}_r)\}}\big) - c\big(S^*_{D_s}\big)\Big\}\Big]. \tag{7}$$

q-ECI gives the expected improvement in coverage score after simultaneously observing the batch of $q$ points $\mathbf{X} = \{\mathbf{x}_1, \ldots, \mathbf{x}_q\}$. However, the resulting Monte Carlo expectation would require $O(q \times m)$ evaluations of Algorithm 1. We therefore adopt a more approximate batching strategy for practical use with large $q$ (see Appendix C for details).

### 3.2.4 `MOCOBO` with Trust Regions

In order to find a set of $K$ solutions, as explained in Section 2, `MOCOBO` follows the `TuRBO`-$M$ principle. That is, it maintains $K$ simultaneous local optimization runs using $K$ individual trust regions. Each local run $k \in \{1, \ldots, K\}$ aims to find a single solution $\mathbf{x}^*_k$, which together form the desired set $S^*$. As in the original `TuRBO` paper, trust regions are rectangular regions of the search space $\mathcal{T}_k \subseteq \mathcal{X}$ defined solely by their size and center point. We center the $K$ trust regions on the best covering set of solutions observed so far during optimization. In particular, on each optimization step $s$, we construct an approximation of $S^*_{D_s}$ using Algorithm 1, and center each trust region $\mathcal{T}_k$ on the corresponding point $\mathbf{x}^{*(s)}_k$ in $S^*_{D_s} = \big\{\mathbf{x}^{*(s)}_1, \ldots, \mathbf{x}^{*(s)}_K\big\}$. We then use our proposed ECI (or q-ECI when $q > 1$) acquisition function to select $q$ candidates for evaluation from within each trust region. The MC approximation of ECI described in Section 3.2.1 can be straightforwardly applied to select candidates in trust region $\mathcal{T}_k$ by sampling the $m$ discrete points from within the rectangular bounds of $\mathcal{T}_k$. Since we select and evaluate $q$ candidates from each of the $K$ trust regions, the total number of observed data points $n$ at step $s$ of `MOCOBO` is $n = s \times K \times q$. As in the original `TuRBO` algorithm, each trust region $\mathcal{T}_k$ has success and failure counters that dictate the size of the trust region. For `MOCOBO`, we count a success for trust region $\mathcal{T}_k$ whenever $\mathcal{T}_k$ proposes a candidate on step $s$ that improves upon the best coverage score and is included in $S^*_{D_{s+1}}$.

## 4 Experiments

We evaluate `MOCOBO` on four high-dimensional, multi-objective BO tasks for which finding a set of $K < T$ solutions to cover the $T$ objectives is desirable. Detailed descriptions of each task are in Section 4.1. Two tasks involve continuous search spaces, allowing direct application of `MOCOBO`, while the other two involve structured spaces (molecules and peptides), requiring an extension for structured optimization.

**Implementation details and hyperparameters.** We implement `MOCOBO` using BoTorch [33] and GPyTorch [34]. Code to reproduce `MOCOBO` results on all tasks considered is available on GitHub: `https://github.com/nataliemaus/mocobo`. We use an acquisition batch size of 20 for all tasks and across all BO methods compared. Since we consider challenging high-dimensional tasks that require a large number of function evaluations, we use approximate GP surrogate models, specifically PPGPR [35]. Further implementation details are provided in Appendix G.

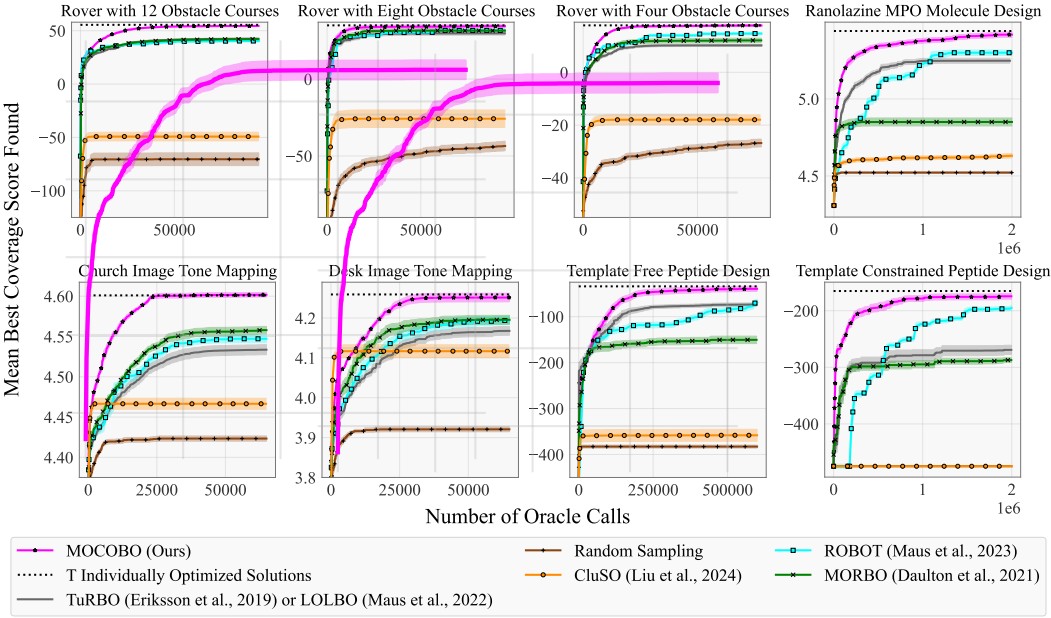

Figure 2: Coverage optimization results on all tasks considered.

**Structured Search Spaces.** Some of our optimization tasks operate on structured search spaces that are not Euclidean. For this, it is typical to employ a generative model—such as a variational autoencoder (VAE; 36, 37)—to convert structured inputs into a continuous latent space for BO [15, 38–40, 40–48], resulting in an algorithm known as *latent space BO*. Naturally, for problems with structured search spaces, we apply `MOCOBO` to the latent space of a pre-trained VAE on which we perform regular end-to-end updates with the surrogate model during optimization [47].

**Plots.** In Figure 2, we plot the best coverage score $c$ Equation (2) obtained by the $K$ best covering solutions found so far after a certain number of function evaluations on each task. Since BO baseline methods are not designed to optimize coverage directly, instead aiming to find a single solution or a set of solutions to optimize the $T$ objectives, we plot the best coverage score $c$ obtained by the $K$ best covering solutions found by the method. All plots show mean coverage scores averaged over 20 replications of each method, and show standard errors.

**Baselines.** In all plots, we compare `MOCOBO` against `CluSO`, `TuRBO`, `ROBOT`, and `MORBO` [14, 20, 21, 28]. While `MORBO` is not designed to optimize coverage, the MOBO setting is closely related to our problem setting, and `MORBO` represents a strong baseline among existing MOBO techniques. We also compare to sampling uniformly at random in the search space. For each method, we compare to the performance of the best set of $K$ solutions found.

**Extending baselines to the structured BO setting.** In the case of the two structured optimization tasks (molecule and peptide design), we apply `LOL-BO` and `ROBOT` as described by design for these problem settings. For `MORBO` and `CluSO`, we take the straightforward approach of applying the methods directly in the continuous latent space of the VAE model without other adaptations. The same pre-trained VAE model is used across methods compared.

Table 1: **Best set of $K = 4$ peptides found by one run of `MOCOBO` for the "template free" peptide design task.** We provide the APEX model's predicted MIC for each sequence on each of the 11 target pathogenic bacteria. The target pathogenic bacteria B1, . . . , B11 are listed in Table H.3. (-) and (+) indicate Gram negative and Gram positive pathogenic bacteria respectively. The best/lowest MIC achieved for each pathogenic bacteria is in bold in each column. See row "TF1" in Figure 3, and Figure B.1 for *in vitro* MICs for this set of $K = 4$ peptides. See Table B.1 for an analogous result on the "template constrained" peptide design task.

| Peptide Amino Acid Sequence | B1(-) | B2(-) | B3(-) | B4(-) | B5(-) | B6(-) | B7(-) | B8(+) | B9(+) | B10(+) | B11(+) |
|---|---|---|---|---|---|---|---|---|---|---|---|
| KKKKLKLKKLKKLLKLLKRL | 1.017 | 1.040 | 1.893 | **0.999** | 8.613 | **0.966** | **1.039** | 65.999 | 38.361 | 338.692 | 1.393 |
| IFHLKILIKILRLL | 0.999 | 15.565 | 1.860 | 1.952 | 404.254 | 486.860 | 406.034 | **1.233** | **1.318** | 7.359 | **0.981** |
| SKKIKLLGLALKLLKLKLKL | 2.654 | 3.268 | 3.113 | 4.854 | **4.923** | 12.967 | 14.610 | 22.631 | 29.685 | 254.306 | 3.947 |
| KKKKLKLKKLKRLLKLKLRL | **0.939** | **0.906** | **1.124** | 1.310 | 10.909 | 1.384 | 1.711 | 12.776 | 32.884 | 434.193 | 1.037 |

**Extending baselines to the coverage optimization setting.** As `TuRBO`, `LOL-BO`, and `ROBOT` target single-objective optimization, we conduct $T$ independent runs to optimize each of the $T$ objectives for the multi-objective task. We use all solutions gathered from the $T$ runs to compute the best covering set of $K$ solutions found by each method. Unlike `LOL-BO` and `TuRBO`, which seek a single best solution for a given objective, a single run of `ROBOT` seeks a set of $M$ solutions that are pairwise diverse. We run `ROBOT` with $M = K$ so that each independent run of `ROBOT` seeks $M = K$ diverse solutions. The aggregate result of the $T$ independent runs for `ROBOT` is thus $K$ diverse solutions for each of the $T$ objectives. We compare to the best covering $K$ solutions from among those $K * T$ solutions. See Appendix G.4 for more details on the diversity constraints used by `ROBOT` and the associated hyperparameters. `MORBO` is a multi-objective optimization method and can thus be applied directly to each multi-objective optimization task. For each run of `MORBO`, we compare to the best covering set of $K$ solutions found among all solutions proposed by the run. For `CluSO`, no extension is needed as this method is designed for coverage optimization. We run `CluSO` with the same $K$ used by `MOCOBO`, and all other hyperparameters set to `CluSO` defaults.

As far as we are aware, `CluSO` is the only existing method that tries to solve the black-box coverage optimization problem directly. We note that other adapted baselines are included not to criticize, but to underscore the importance of explicitly addressing this problem class.

**T individually optimized solutions baseline.** We also compare to a brute-force method involving $T$ separate single-objective optimizations for each of the $T$ objectives, using `TuRBO` for each run or `LOL-BO` for molecule and peptide design. This is not an alternative for finding $K < T$ covering solutions, but instead identifies $T$ solutions, one per objective, approximating a ceiling we can achieve on performance without the limit of $K < T$ solutions. Approaching the performance of this baseline implies that we can find $K$ solutions that do nearly as well as if we were allowed $T$ solutions instead.

### 4.1 Tasks

**Peptide design.** In the peptide design task, we explore amino acid sequences to minimize the MIC (minimum inhibitory concentration, measured in $\mu\mathrm{mol}\,L^{-1}$) for each of $T = 11$ target drug resistant strains or otherwise challenging to kill bacteria (B1-B7 Gram negative, B8-B11 Gram positive). Table H.3 lists our target bacteria in this study. Briefly, MIC indicates the concentration of peptide needed to inhibit bacterial growth (see Kowalska-Krochmal and Dudek-Wicher 49). We evaluate MIC for a given peptide sequence and bacteria using the APEX 1.1 model proposed by Wan et al. [50]. To frame the problem as maximization, we optimize $-$MIC. We seek $K = 4$ peptides that together form a potent set of antibiotics for all $T = 11$ bacteria. To enable optimization over peptides, we use the VAE model pre-trained on $4.5$ million amino acid sequences from Torres et al. [23] to map the peptide sequence search space to a continuous 256 dimensional space.

**Template free vs template constrained peptide design.** We evaluate `MOCOBO` on two variations of the peptide design task: "template free" (TF) and "template constrained" (TC). For TF, we allow the optimizer to propose any sequence of amino acids. For TC, we add a constraint that any sequence proposed by the optimizer must have a minimum of 75 percent sequence similarity to at least one of the 10 template amino acid sequences in Table H.4. These 10 templates were mined from extinct organisms and selected by Wan et al. [50]. The motivation of the template constrained task is to design peptides specifically likely to evade antibiotic resistance by producing "extinct-like" peptides that bacteria have not encountered in nature in thousands of years. For `MOCOBO` and all BO baselines (`TuRBO`, `LOL-BO`, `ROBOT`, and `MORBO`), we handle the optimization constraint by adapting techniques from `SCBO` [51]. For `CluSO` and random sampling, we handle the constraint with rejection sampling.

**Ranolazine MPO molecule design.** Ranolazine is a drug used to treat chest pain. The original `Ranolazine MPO` task from the Guacamol benchmark suite of molecular design tasks [52] aims to design an alternative to this drug: a molecule with a high fingerprint similarity to Ranolazine that includes fluorine. We extend this task to the multi-objective optimization setting by searching for alternatives to Ranolazine that include $T = 6$ reactive nonmetal elements not found in Ranolazine: fluorine, chlorine, bromine, selenium, sulfur, and phosphorus. We aim to cover the $T = 6$ objectives with $K = 3$ molecules. We use the SELFIES-VAE introduced by Maus et al. [47] to map the molecular space to a continuous 256 dimensional space.

**Rover.** The rover optimization task introduced by Wang et al. [53] consists of finding a 60-dimensional policy that allows a rover to move along some trajectory while avoiding a fixed set of obstacles. To frame this as a multi-objective optimization task, we design $T$ unique obstacle courses

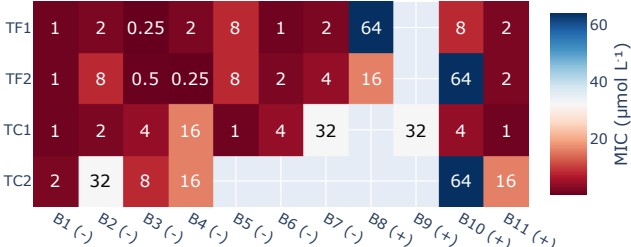

Figure 3: *In vitro* results for the two best "template free" (TF1, TF2) and two best "template constrained" (TC1, TC2) runs of `MOCOBO` for the peptide design task. Columns are the best/lowest *in vitro* MIC among the $K = 4$ peptides found by `MOCOBO` for each target pathogenic bacteria B1, . . . , B11 listed in Table H.3. (-) and (+) indicate Gram negative and Gram positive respectively. TF1 and TC1 correspond to the single runs of `MOCOBO` shown in Table 1 and Table B.1 respectively. Methods used to obtain *in vitro* MICs are provided in Appendix A.

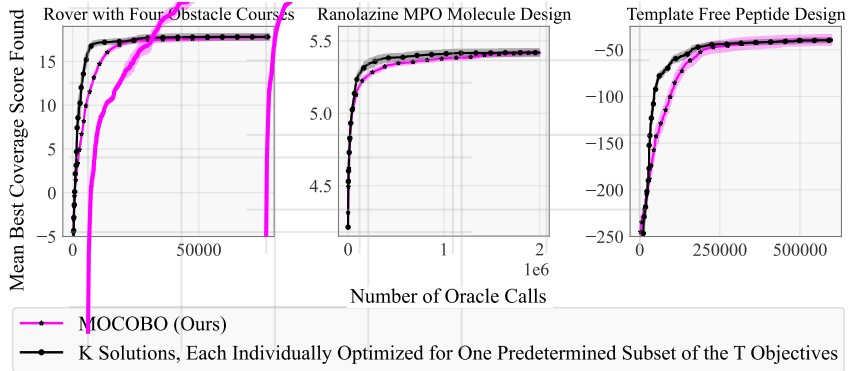

Figure 4: Ablation study comparing `MOCOBO` to optimization performance where a known "good" partitioning of the $T$ objectives into $K$ subsets is available in advance. We individually optimize $K$ solutions, one for each partition.

for the rover to navigate. The obstacle courses are designed such that no single policy can successfully navigate all courses. We seek $K < T$ policies so that at least one policy enables the rover to avoid obstacles in each course. We evaluate on three unique instances with varying numbers of obstacle courses. For the instances of this task with $T = 4$ and $T = 8$, we seek to cover the objectives with $K = 2$ solutions. For the $T = 12$ instance, we seek to cover the objectives with $K = 4$ solutions.

**Image tone mapping.** In high dynamic range (HDR) images, some pixels (often associated with light sources) can dominate overall contrast, requiring adjustments to reveal detail in low-contrast areas, a problem known as *tone mapping* [54, Section 6]. Tone mapping algorithms involve various tunable parameters, resulting in a high-dimensional optimization problem of subjectively perceived quality. We seek a covering set of $K = 4$ solutions to optimize a set of $T = 7$ image aesthetic (IAA) and quality (IQA) assessment metrics from the `pyiqa` library [55] (see metrics listed in Table H.1). Our practical goal is that, while we do not know *a priori* which metric is best for a particular image, covering all metrics may result in at least one high quality image. We optimize over the 13-dimensional parameter space of an established tone mapping pipeline to tone-map the "Stanford Memorial Church" [56] and "desk lamp" [57] benchmark images. See Appendix H.1 for details.

### 4.2 Optimization Results

In Figure 2, we provide optimization results comparing `MOCOBO` to the baselines discussed above on all tasks. The results show that `MOCOBO` finds sets of $K$ solutions that achieve higher coverage scores across tasks. The "T Individually Optimized Solutions" baseline appears as a horizontal dotted line in all plots of Figure 2, representing the average coverage score of $T$ individually optimized solutions, serving as an approximation of the best possible performance *without* the constraint of a limited $K < T$ solution set. Results in Figure 2 demonstrate that the smaller set of $K < T$ solutions identified by `MOCOBO` nearly equals the performance of the complete set of $T$ individually optimized solutions. This result depends on using domain knowledge to choose $K$ large enough to achieve it.

Results in Figure 2 show that matching the performance of $T$ optimized solutions is possible with some values of $K \ll T$.

Although CluSO exhibits substantially lower performance than other baselines across the tasks we consider—including baselines not explicitly designed for the coverage problem—this should not be viewed as a criticism of the method. Rather, it reflects that CluSO was developed for low-dimensional black-box problems and does not scale effectively to the high-dimensional and structured optimization settings we study, further motivating the need for our scalable MOCOBO approach.

**Peptide design results.** In Table 1, we provide the $K = 4$ peptides found by one run of MOCOBO for the template free (TF) peptide design task. For each peptide, we provide the the APEX 1.1 model's predicted MIC for each of the 11 target bacteria. MIC values $\leq 16 \, \mu\mathrm{mol}\,\mathrm{L}^{-1}$ are considered to be "highly active" against the target pathogenic bacteria [50]. We highlight in Table 1 the comparison between the second peptide with the other three peptides. B1-B7 are Gram negative (GN) bacteria, while B8-B11 are Gram positive (GP). Peptide 2 specialized to the GP bacteria (B8-B11 scores predicted highly active compared to the other peptides) at the expense of broad spectrum activity for the GN bacteria (B5, B6, B7 predicted inactive). By specializing its solutions to GN and GP bacteria separately, MOCOBO achieves low MIC across the target bacteria with only $K = 4$ peptides. A similar table of results is also provided for the template constrained (TC) peptide design task (see Table B.1).

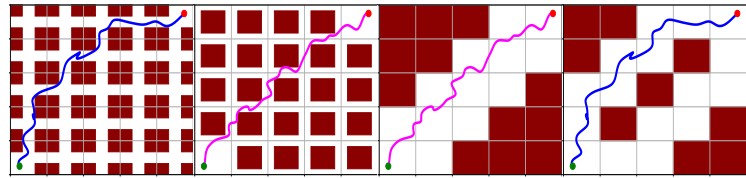

Figure 5: $T = 4$ varied obstacle courses as multiple objectives for the rover task and $K = 2$ covering trajectories (blue, magenta) found by MOCOBO. The first trajectory (magenta) successfully navigates obstacle courses 2 and 3 (second and third panels from the left). The second trajectory (blue) navigates obstacle courses 1 and 4 (first and fourth panels from the left).

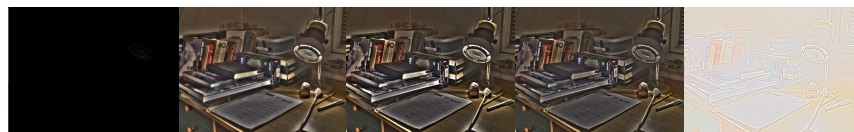

Figure 6: **(Leftmost Panel)** Initial hdr desk image for the desk variation of the "image tone mapping" task. **(Four Rightmost Panels)** Images obtained by transforming the hdr desk image using the best covering set of $K = 4$ solutions found by a single run of MOCOBO.

In Figure 3, we provide *in vitro* results for the two best TF and two best TC runs of MOCOBO for the peptide design task. Here, "best" means runs that achieved highest coverage scores according to the APEX 1.1 model. For each run, Figure 3 provides the best/lowest *in vitro* MIC among the $K = 4$ peptides found by MOCOBO for each target bacteria. Results demonstrate that solutions found by MOCOBO optimizing against the *in-silico* APEX 1.1 model achieve good coverage of the 11 target pathogenic bacteria *in vitro*. In Figure B.1, we provide the full set of *in vitro* results for each these runs of MOCOBO, with MIC values obtained by each of the $K = 4$ peptides found for each target bacteria. Methods used to obtain *in vitro* MICs are provided in Appendix A.

**Molecule design results.** In Table B.2, we provide results for the $K = 3$ molecules found by one run of MOCOBO for the molecule design task. Each of the 6 target elements is successfully present in one of the $K = 3$ molecules in the best covering set. These three molecules therefore effectively cover the $T = 6$ objectives, with each objective having a max score $> 0.9$.

**Rover results.** In Figure 5, we depict the 4 varied obstacle courses used for the $T = 4$ variation of the rover task and $K = 2$ covering trajectories found by a single run of MOCOBO. In Figure B.3 and B.4, we provide analogous figures for the $T = 8$ and $T = 12$ variations. In each example, the MOCOBO optimized set of trajectories covers all obstacle courses such that all obstacles are avoided.

**Image tone mapping results.** In Figure 6 and Figure B.2, we provide the original HDR images and the images obtained by the $K = 4$ solutions found by a single run of MOCOBO for the church and desk test images, respectively. In both variations, three of the four solutions result in adequately

tone-mapped images, while one results in a poor-quality image (see the middle-right church image and the rightmost desk image). The poor quality images were each generated by the one solution that MOCOBO used to cover metrics 5 and 6 (see metric ID numbers in Table H.1). This indicates that metrics 5 and 6 are poor indicators of true image quality in this setting. This result highlights a useful application of MOCOBO: By dedicating one of the $K$ solutions to maximizing the misleading metrics (5 and 6), the other three solutions to can focus on the remaining subjectively better metrics.

### 4.3 Ablation study

A challenging aspect of our problem setting is that we do not know *a priori* the best way(s) to divide the $T$ objectives into $K$ subsets such that good coverage can be obtained. In this section, we seek to answer the question: *what is the efficiency lost by MOCOBO due to not knowing an efficient partitioning of the objectives in advance?* To construct a proxy "efficient" partition to measure this, we first run MOCOBO to completion on a task, and then consider optimization as if we had known the partitioning of objectives found at the end in advance.

With such an "oracle" partitioning in hand ahead of time, we can efficiently optimize by running $K$ independent optimization runs: one to find a single solution for each subset in the fixed, given partition. With a fixed partition, the coverage score in (2) reduces to optimizing the sum of objectives in each relevant subset independently. In Figure 4, we compare this strategy directly to MOCOBO on three optimization tasks. We plot the best coverage score obtained by $K$ independent runs of TuRBO/LOL-BO on the fixed oracle $K$-partitioning of the objectives by combining their solutions into a single set. MOCOBO achieves the same average best coverage scores with minimal loss in optimization efficiency *despite having to discover an efficient partitioning during optimization.* See Appendix B.2 for additional ablation studies.

## 5 Related Works

The coverage optimization problem setting considered in this paper has not been previously explored in the Bayesian optimization literature. However, similar problem settings [17–19] have been studied in the context of *gradient*-based multi-objective optimization. For instance, Ding et al. [17] demonstrated that the coverage problem generalizes various clustering methods, including k-means clustering, and proposed an algorithm that extends k-means++ [58] and Lloyd's algorithm [59] to efficiently solve the coverage problem using gradient descent. Li et al. [18] introduced Many-objective Multi-solution Transport (MosT), a novel framework that scales multi-objective gradient-based optimization to a large number of objectives. MosT uses an optimal transport to establish a balanced assignment between objectives and solutions, allowing each solution to focus on a specific subset of objectives and ensuring that the collective set of solutions covers all objectives. Lin et al. [19] introduced a novel Tchebycheff set (TCH-Set) scalarization approach for covering multiple conflicting objectives using a small set of collaborative solutions. They further propose a smooth Tchebycheff set (STCH-Set) scalarization method to handle non-smoothness in TCH-Set scalarization. The reliance on gradients makes these previous approaches inapplicable to the black-box setting.

For the black-box setting, Liu et al. [20] was the first to consider the coverage problem. They propose an analogous formalization of the coverage problem in the context of multi-objective black-box optimization, along with a clustering-based swarm optimization algorithm (CluSO) designed to solve it. In Section 4, we provide empirical results demonstrating that our Bayesian optimization approach (MOCOBO) consistently outperforms CluSO by a large margin across all tasks considered.

Lastly, we note that Malkomes et al. [29] proposed a method with the same name as the ECI we defined in Section 3, which was later extended in [60]. The resemblance is only in the name, as their work addresses a fundamentally different notion of coverage. Mainly, they attempt to cover the *input space* by identifying a diverse set of feasible solutions that span a subspace defined by known threshold constraints on each objective. In contrast, we attempt to "cover" the different *objectives*.

## 6 Conclusions

By bridging the gap between traditional multi-objective Bayesian optimization (BO) and practical coverage requirements, MOCOBO offers an effective approach to tackle critical problems in drug design and beyond. This framework extends the reach of Bayesian optimization into new domains, providing a robust solution to the challenges posed by extreme trade-offs and the need for specialized, collaborative solutions. See Appendix D for additional discussion and limitations of MOCOBO.

## Acknowledgments and Disclosure of Funding

N. Maus was supported by the National Science Foundation Graduate Research Fellowship; K. Kim was supported by a gift from AWS AI to Penn Engineering's ASSET Center for Trustworthy AI; J. R. Gardner was supported by NSF awards IIS-2145644 and DBI-2400135; C. de la Fuente-Nunez was supported by NIH grant R35GM138201 and Defense Threat Reduction Agency grants HDTRA11810041, HDTRA1-21-1-0014, and HDTRA1-23-1-0001.

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

# Contents

# A   Obtaining *In Vitro* Minimal Inhibitory Concentration (MIC) Data

In this section, we provide the methods used to produce all *in vitro* minimal inhibitory concentration (MIC) values reported in this paper.

**Peptide synthesis.**   All peptides were synthesized by solid-phase peptide synthesis using the Fmoc strategy and purchased from AAPPTec.

**Bacterial strains and growth conditions used in the experiments.**   The following Gram-negative bacteria were used in our study: Acinetobacter baumannii ATCC 19606, Escherichia coli ATCC 11775, E. coli AIC221 (E. coli MG1655 phn$E_2$::FRT), E. coli AIC222 (E. coli MG1655 pmrA53 phn$E_2$::FRT (colistin resistant)), Klebsiella pneumoniae ATCC 13883, Pseudomonas aeruginosa PAO1, and P. aeruginosa PA14. The following Gram-positive bacteria were also used in our study: Staphylococcus aureus ATCC 12600, S. aureus ATCC BAA-1556 (methicillin-resistant strain), Enetrococcus faecalis ATCC 700802 (vancomycin-resistant strain) and E. faecium ATCC 700221 (vancomycin-resistant strain). Bacteria were grown from frozen stocks and plated on Luria-Bertani (LB) or Pseudomonas isolation agar plates (P. aeruginosa strains) and incubated overnight at $37\,^\circ$C. After the incubation period, a single colony was transferred to $6\,\mathrm{mL}$ of LB medium, and cultures were incubated overnight ($16\,\mathrm{h}$) at $37\,^\circ$C. The following day, an inoculum was prepared by diluting the overnight cultures $1:100$ in $6\,\mathrm{mL}$ of the respective media and incubating them at $37\,^\circ$C until bacteria reached logarithmic phase ($OD_{600} = 0.3 - 0.5$).

**Antibacterial assays.**   The *in vitro* antimicrobial activity of the peptides was assessed by using the broth microdilution assay [61]. Minimal inhibitory concentration (MIC) values of the peptides were determined with an initial inoculum of $2 \times 10^6\,\mathrm{cells\,mL^{-1}}$ in LB in microtiter 96-well flat-bottom transparent plates. Aqueous solutions of the peptides were added to the plate at concentrations ranging from $0.0625$ to $64\,\mathrm{\mu mol\,L^{-1}}$. The lowest concentration of peptide that inhibited 100 percent of the visible growth of bacteria was established as the MIC value in an experiment of $20\,\mathrm{h}$ of exposure at $37\,^\circ$C. The optical density of the plates was measured at $600\,\mathrm{nm}$ using a spectrophotometer. All assays were done as three biological replicates.

# B Additional Results

In this section, we provide all additional empirical results not included in the main text.

## B.1 Additional Covering Sets of Solutions Found by `MOCOBO`

In this section, we provide additional examples of covering sets of solutions found by `MOCOBO` for various tasks from Section 4.

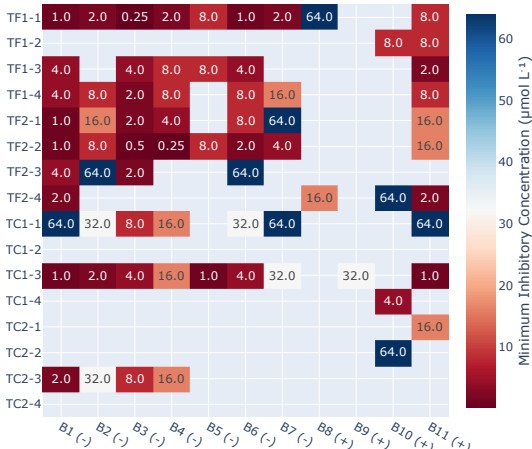

Figure B.1: *In vitro* results for the two best "template free" (TF1, TF2) and two best "template constrained" (TC1, TC2) runs of `MOCOBO` for the peptide design task. Here, "best" means runs that achieved highest coverage scores according to the APEX 1.1 model. Each row is a single peptide found by a run of `MOCOBO`. Row TF$i$-$j$ indicates template free run $i$, peptide $j$. Similarly, TC$i$-$j$ indicates template constrained run $i$, peptide $j$. Columns are the *in vitro* MICs for each target pathogenic bacteria B1, ..., B11 listed in Table H.3. (-) and (+) indicate Gram negative and Gram positive respectively. TF1 and TC1 correspond to the single runs of `MOCOBO` shown in Table 1 and Table B.1 respectively. Methods used to obtain *in vitro* MICs are provided in Appendix A.

In Figure B.1, we provide *In vitro* results for the two best "template free" and two best "template constrained" runs of `MOCOBO` for the peptide design task. Here, "best" means runs that achieved highest coverage scores according to the APEX 1.1 model. Figure B.1 provides *in vitro* MICs for each of the $K = 4$ peptides found by each of these runs of `MOCOBO`, for each of the 11 target pathogenic bacteria. Methods used to obtain *in vitro* MICs are provided in Appendix A.

Table B.1: The best set of $K = 4$ peptide sequences found by one run of `MOCOBO` for the "template free" peptide design task described in Section 4.1. For each of the four sequences, we provide the MIC according to the APEX model for each of the 11 target pathogenic bacteria. The target pathogenic bacteria B1, ..., B11 are listed in Table H.3. (-) and (+) indicate Gram negative and Gram positive pathogenic bacteria respectively. The best/lowest MIC achieved for each pathogenic bacteria is in bold in each column. See row "TC1" in Figure 3, and Figure B.1 for *in vitro* MICs for this set of $K = 4$ peptides.

| Peptide Amino Acid Sequence | B1(-) | B2(-) | B3(-) | B4(-) | B5(-) | B6(-) | B7(-) | B8(+) | B9(+) | B10(+) | B11(+) |
|---|---|---|---|---|---|---|---|---|---|---|---|
| KKLKIIRLLFK | 18.594 | 17.067 | 4.278 | 5.352 | **13.460** | 50.442 | 24.543 | 456.831 | 431.276 | 441.292 | 20.305 |
| WAIRGLKLATWLSLNNKF | 6.771 | 20.358 | 14.644 | 10.477 | 65.172 | 97.404 | 59.195 | **19.846** | **33.459** | 237.697 | 7.708 |
| RWARNLVRYVKWLKKLKKVI | **2.171** | **4.589** | **2.641** | **3.073** | 54.400 | **11.444** | **19.150** | 75.588 | 89.977 | 413.386 | **2.913** |
| HWITIAFFRLSISLKI | 225.260 | 346.589 | 56.583 | 58.253 | 458.963 | 475.616 | 538.352 | 293.852 | 338.047 | **34.230** | 22.153 |

In Table B.1, we provide an example of a covering set of peptides found by `MOCOBO` for the "template constrained" variation of the peptide design task.

In Figure B.2, we provide the original HDR image (leftmost panel), and the $K = 4$ images produced using the $K = 4$ solutions found by a single run of `MOCOBO` for the church image variation of the

image tone mapping task. An analogous result for the desk image variation of the image tone mapping task can be found in the main text in Figure 6. A notable apparent limitation of the existing IAA/IQA metrics we used for the image tone mapping tasks is that they all favored monochromatic images for both the church and desk images. As such, the tone-mapped images obtained appear less colorful than their hand-tuned counterparts reported in prior work.

In Table B.2, we provide an example of a covering set of $K = 3$ molecules found by MOCOBO for the molecule design task. As mentioned in Section 4, these three molecules effectively cover the $T = 6$ objectives, as evidenced by the presence of all $T = 6$ target elements in one of the $K = 3$ molecules designed by MOCOBO.

In Figure B.3 and Figure B.4, we provide examples of a covering set of trajectories found for the $T = 8$ and $T = 12$ variations of the rover task respectively. In each figure, a panel is shown for each of the $T$ obstacles courses, with obstacles colored in red. The required starting point for the rover is a green point in the bottom left of each panel. The "goal" end point that the rover aims to reach without hitting any obstacles is the red point in the top right of each panel. Each panel also shows the best among the $K$ trajectories found by a run of MOCOBO for navigating each obstacle course. An analogous plot for the $T = 4$ variation of the rover optimization task is provided in the main text in Figure 5.

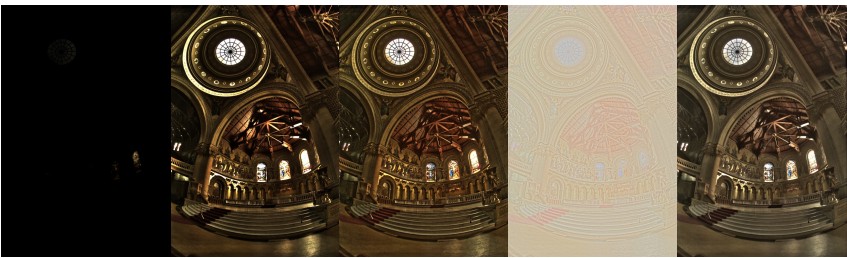

Figure B.2: **(Leftmost Panel)** Image obtained by naively compressing the dynamic range of the HDR church image in the "image tone mapping" task. **(Four Rightmost Panels)** Images obtained by applying tone mapping to the church image using the best covering set of $K = 4$ solutions found by a single run of MOCOBO.

Table B.2: The best set of $K = 3$ molecules found by one run of MOCOBO for the Ranolazine MPO multiple element molecule design task described in Section 4.1. For each of the three molecules, we provide the objective value obtained for each of the $T = 6$ objectives. The best/highest objective value is in bold in each column. Each objective aims to add a different target element to Ranolazine (F, Cl, Br, Se, S, and P). The $T = 6$ target elements are bold and colored blue in the SMILES string [62] representation of each molecule.

| Molecule (SMILES String) | Obj 1 (add F) | Obj 2 (add Cl) | Obj 3 (add Br) | Obj 4 (add Se) | Obj 5 (add S) | Obj 6 (add P) |
|---|---|---|---|---|---|---|
| CC=C(C)C(OC(=O)C(O)CCCCCC(=O)O)
=CC=CCCCCCC[**Se**]CC(=O)NC1=CC=CC=C1C | 0.8038 | 0.8038 | 0.8038 | **0.9108** | 0.8038 | 0.8038 |
| CC=C(C)C(OC(=O)CCCCCC(O)C(=**S**)**Cl**)
=CC=CCOCCCCC(O)CC(=O)NC1=CC=CC=C1C | 0.8043 | **0.9114** | 0.8043 | 0.8043 | **0.9114** | 0.8043 |
| CC=C(C)C(OC(=O)C(O)CCCCCC(=O)C**Br**)
=CC=COC**P**CCCCN(C)CC(=O)[NH1]C1=CC=C(**F**)C=C1C | **0.9097** | 0.8028 | **0.9097** | 0.8028 | 0.8028 | **0.9097** |

## B.2 Additional Ablation Studies

In this section, we provide additional ablation studies not included in the main text.

### B.2.1 Ablation: Settings Where All Objectives are Highly Conflicting

MOCOBO is designed for cases where some, but not all, of the objectives are highly conflicting, and this is the case for all of the tasks considered in Section 4. The conflicting pairs of the objectives prevent a single solution from optimizing all objectives well. The fact that not *all* pairs of objectives are completely conflicting, is the reason why it is possible to cover all T objectives with a small set of

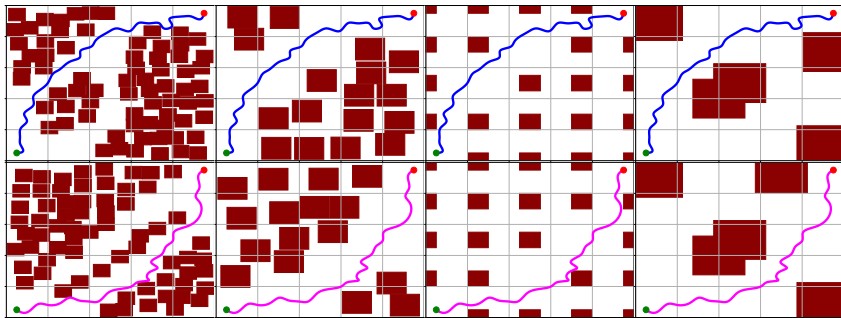

Figure B.3: The 8 panels depict the 8 obstacle courses that the rover must navigate for the $T = 8$ variation of the rover task, with obstacles colored in red. The required starting point for the rover is a green point in the bottom left of each panel. The "goal" end point that the rover aims to reach without hitting any obstacles is the red point in the top right of each panel. The line in each panel shows the best trajectory for navigating the obstacle course from among the $K = 2$ covering trajectories found by a single run of MOCOBO. The first trajectory in the covering set is shown in magenta and successfully navigates obstacle courses 5, 6, 7, and 8 **(Bottom Row)**. The second is shown in blue and successfully navigates obstacle courses 1, 2, 3, and 4 **(Top Row)**.

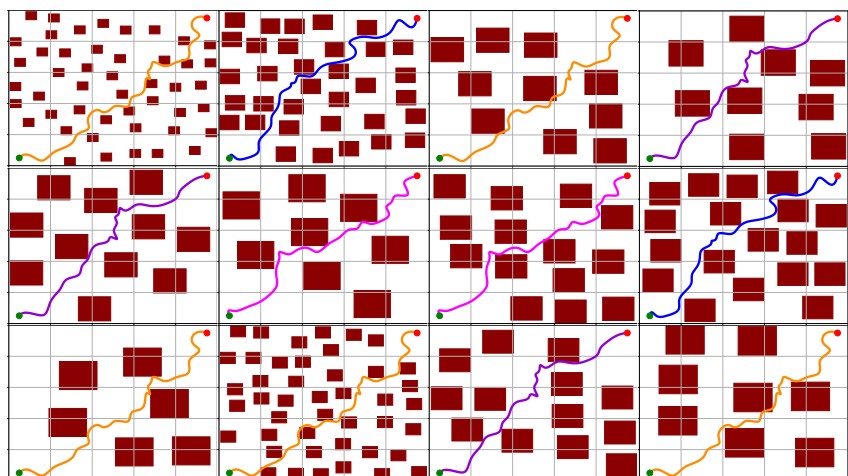

Figure B.4: The 12 panels depict the 12 obstacle courses that the rover must navigate for the $T = 12$ variation of the rover task, with obstacles colored in red. The required starting point for the rover is a green point in the bottom left of each panel. The "goal" end point that the rover aims to reach without hitting any obstacles is the red point in the top right of each panel. The line in each panel shows the best trajectory for navigating each obstacle course from among the $K = 4$ covering trajectories found by a run of MOCOBO. The first trajectory in the covering set is shown in magenta and successfully navigates obstacle courses 6 and 7 **(Middle Row: Center Left and Center Right Panels)**. The second is shown in blue and successfully navigates obstacle courses 2 and 8 **(Top Row: Center Left Panel, Middle Row: Rightmost Panel)**. The third is shown in purple and successfully navigates obstacle courses 4, 5, and 11 **(Top Row: Rightmost Panel, Middle Row: Leftmost Panel, and Bottom Row: Center Right Panel)**. The fourth trajectory is shown in orange and successfully navigates obstacle courses 1, 3, 9, 10, and 12 **(Top Row: Leftmost Panel and Center Right Panel, Bottom Row: Leftmost, Center Left, and Rightmost Panels)**.

K solutions. If all pairs of the T objectives are highly conflicting, there is by definition no possible set of $K < T$ solutions such that all objectives are well optimized. In this case, it's best to use T individually optimized solutions if your goal is to find at least one solution that well optimizes each objective. However, it is still interesting to consider the performance of MOCOBO in the setting of

T objectives that are pairwise highly conflicting. To investigate this, we construct a new variation of the multi-objective rover task described in Section 4.1. We design T=3 obstacle courses such that it is impossible for the rover to take any single path that avoids all obstacles in any pair of the obstacle courses. This results in T=3 objectives that are pairwise highly conflicting. We then run 20 replications of MOCOBO on this problem with K=2, asking MOCOBO to design K=2 solutions that cover the T=3 pairwise highly conflicting objectives. For this task, MOCOBO got an average coverage score of 2.932, while T individually optimized solutions got an average coverage score of 12.899. It is unsurprising that T individually optimized solutions achieved better coverage here since all three objectives are pairwise highly conflicting. However, this result demonstrates that MOCOBO is able to design sets of K=2 solutions that still achieve fairly high coverage scores.

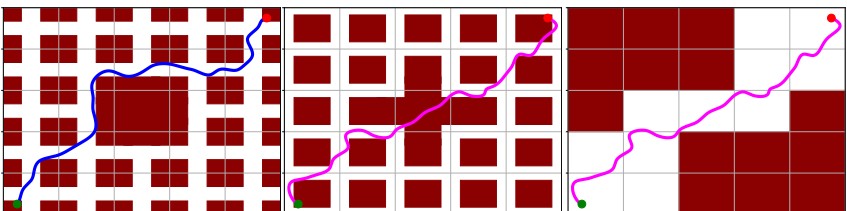

Figure B.5: The three panels depict the T=3 obstacle courses that the rover must navigate for the ablation variation of the rover task where all T=3 objectives pairwise highly conflicting. Obstacles are colored in red. The three obstacle courses are designed such that any single path that successfully avoids all obstacles in one obstacle course, must hit some obstacles in both of the other obstacle courses. The required starting point for the rover is a green point in the bottom left of each panel. The "goal" end point that the rover aims to reach without hitting any obstacles is the red point in the top right of each panel. The line in each panel shows the best trajectory for navigating the obstacle course from among the K=2 covering trajectories found by a single run of MOCOBO. The first trajectory in the covering set is shown in magenta and is the best trajectory for navigating obstacle courses 2 and 3 (**Middle Panel and Rightmost Panels**). The second is shown in blue and successfully navigates obstacle course 1 (**Leftmost Panel**).

Figure B.5 provides a diagram of the T=3 pairwise highly conflicting obstacle courses, and an example of one of the covering sets of K=2 trajectories found by a single run of MOCOBO. One optimized trajectory (shown in blue) is designed by MOCOBO such that it specializes to navigate the first obstacle course, navigating it without hitting any obstacles (Leftmost Panel). The other optimized trajectory (shown in magenta) is designed by MOCOBO such that it avoids all obstacles in the third obstacle course (Rightmost Panel), while also minimizing total amount of impact with the obstacles in the second obstacle course (Middle Panel). Since it is by-design impossible to avoid all obstacles in all T=3 obstacle courses with only K=2 solutions, Figure B.5 demonstrates that MOCOBO was able to successfully balance trade-offs among the objectives, designing a set of K=2 solutions that minimized the total amount of obstacle impact across the three obstacle courses.

### B.2.2   Ablation: MOCOBO Covering Set Size

In this section, we ablate $K$, the user-specified hyperparameter that dictates of size of the set that MOCOBO designs to cover the $T$ objectives. For this ablation, we use the rover task with $T = 4$ obstacle courses as defined in Section 4.1. Note that in the main text we provide results comparing MOCOBO to baseline methods using $K = 2$ for this task. In Figure B.6, we provide results from running MOCOBO with each of $K = 1, 2, 3$, and $4$. With the ability to use more than 2 solutions to cover the $T$ objectives ($K = 3, 4$), the optimization problem becomes easier and MOCOBO is able to converge more quickly. However, the loss in optimization efficiency inured by using the smaller covering set size of $K = 2$, rather than a higher value of $K$, is marginal, highlighting MOCOBO's ability of efficiency design smaller sets of high performing solutions.

With only one solution ($K = 1$), it is by definition not possible to cover all $T = 4$ objectives since several pairs of the obstacle courses are specifically designed to be completely conflicting (no one trajectory can successfully avoid the obstacles in all $T = 4$ obstacle courses). MOCOBO with $K = 1$ therefore obtains a substantially lower final coverage score.

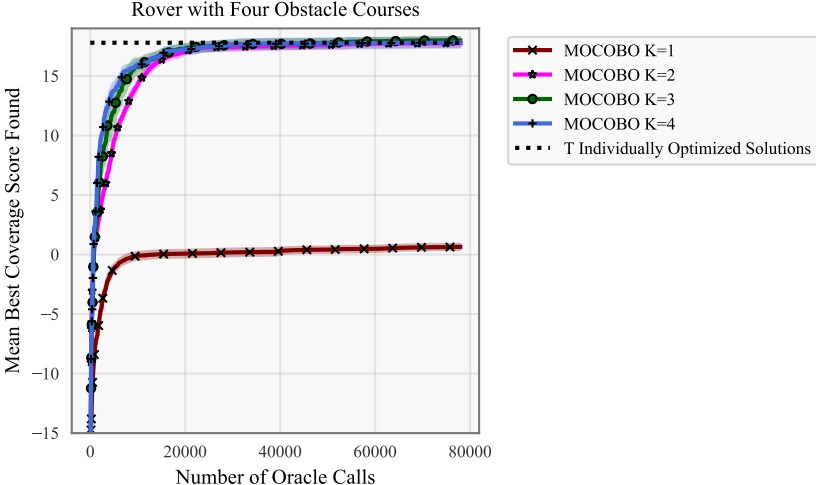

Figure B.6: Ablating the hyperparameter $K$ used to run MOCOBO on the rover task with $T = 4$ obstacle courses (see task definition in Section 4.1).

### B.2.3  Ablation: MOCOBO **Surrogate Model Quality**

In this section, we ablate the quality of the surrogate model used by MOCOBO. To run MOCOBO with surrogate models of varying quality, we vary the MOCOBO hyperparameter $m$: the number of inducing points used to define the approximate Gaussian process (GP) surrogate model (see surrogate model details in Appendix G). It is well established in the literature that approximate GP models perform better with a larger number of inducing points, as the inducing point approximation used by the model is improved. However, there is an inherent trade-off as the computational cost of training the model increases with the number of inducing points. In practice, we therefore often to select the smallest possible value of $m$ (to maximize computational efficiency) such that we don't incur any significant performance degradation.

For this ablation, we use the rover task with $T = 4$ obstacle courses as defined in Section 4.1. In Figure B.7, we provide results from running MOCOBO with each of $m = 4, 16, 64, 256, 1024$, and 4096. In all other experiments in this paper, we use $m = 1024$. Results in Figure B.7 support our choice of $m = 1024$ as results demonstrate that no significant performance improvement is gained by using the larger value of $m = 4096$, and that performance starts to degrade with smaller values of $m \leq 256$.

### B.2.4  Ablation: MOCOBO **Batch Size**

In this section, we ablate the MOCOBO hyperparameter $q$, the acquisition batch size which dictates how many points are selected for evaluation from each trust region on each iteration of MOCOBO. For this ablation, we use the rover task with $T = 4$ obstacle courses as defined in Section 4.1. In Figure B.8, we provide results from running MOCOBO with each of $q = 1, 2, 5, 10, 20$, and 40. In all other experiments in this paper, we use $q = 20$. Results in Figure B.8 demonstrate the robustness MOCOBO to changes in $q$, as there is little to no significant change in the performance of MOCOBO with different the values of $q$.

### B.2.5  Ablation: MOCOBO **Trust Regions**

In this section, we ablate the use of trust regions in MOCOBO. For this ablation, we use the rover task with $T = 4$ obstacle courses as defined in Section 4.1. To ablate the use of trust regions, we compare to MOCOBO (as defined in Section 3 with trust regions), to running MOCOBO without trust regions. Results in Figure B.9 demonstrate that MOCOBO performs significantly better with the use of trust regions. This result confirms that trust regions significantly improve performance in the high-dimensional settings we consider.

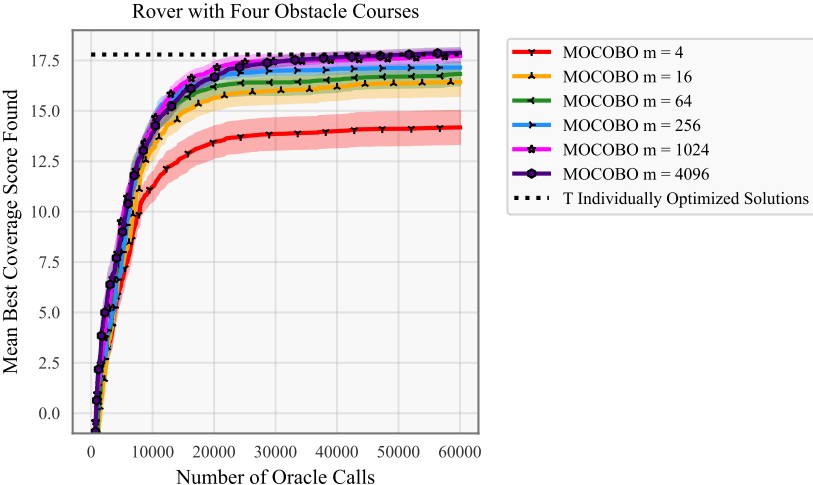

Figure B.7: Ablating the `MOCOBO` hyperparameter $m$ (the number of inducing points used by the surrogate model) on the rover task with $T = 4$ obstacle courses (see task definition in Section 4.1). Lower values of $m$ correspond to lower surrogate model quality.

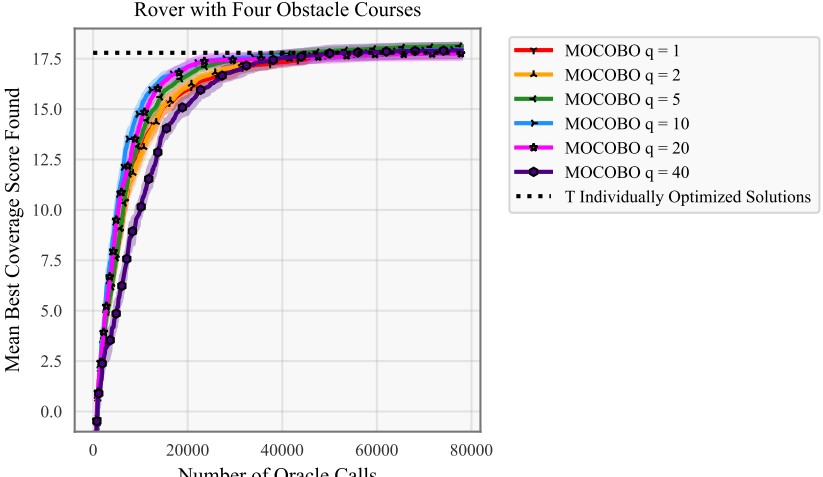

Figure B.8: Ablating the `MOCOBO` hyperparameter $q$ (the number of points selected for evaluation during the acquisition step on each iteration) on the rover task with $T = 4$ obstacle courses (see task definition in Section 4.1).

**Why trust regions**  It has been well-established in the Bayesian optimization (BO) literature that standard BO (without trust regions or other high-dimensional adaptations) performs poorly in high dimensions. Eriksson et al. [21] introduced trust regions as a principled way to improve performance of high-dimensional BO. Since then, trust regions have become a standard tool for any high-dimensional BO task. Since all of the tasks we consider in this paper are high-dimensional, this precisely why we adopt trust regions. We note that recent work has proposed alternative approaches to improve high-dimensional BO without trust regions (e.g., [63]). However, there remain concerns about the applicability of these approaches to structured domains (e.g., [64]).

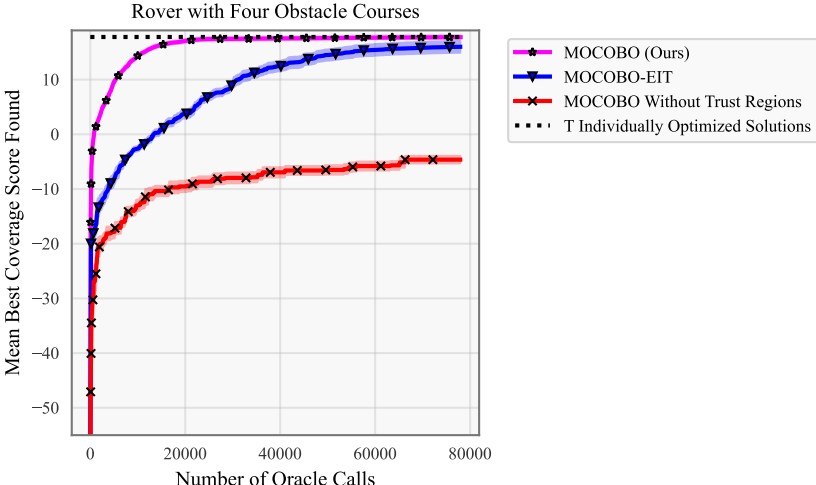

Figure B.9: Ablation comparing `MOCOBO` as defined in Section 3, to two variations each removing a different component of `MOCOBO`. To ablate the use of trust regions in `MOCOBO`, we compare to `MOCOBO` without trust regions. To ablate our proposed ECI acquisition function, we additionally compare to `MOCOBO-EIT`, a variation of `MOCOBO` where we select candidates in acquisition by taking the maximum of standard EI for each of the $T$ objectives individually (EIT), rather than using ECI as described in Section 3. We provide optimization results for `MOCOBO` and these two variations on the rover task with $T = 4$ obstacle courses (see task definition in Section 4.1).

### B.2.6 Ablation: `MOCOBO` Expected Coverage Improvement (ECI) Acquisition Function

In this section, we ablate our proposed expected coverage improvement (ECI) acquisition function. For this ablation, we use the rover task with $T = 4$ obstacle courses as defined in Section 4.1. To ablate the use of ECI, we compare to `MOCOBO` (as defined in Section 3 with ECI), to `MOCOBO-EIT`: a variation of `MOCOBO` where on each iteration, we instead select points for evaluation by taking the maximum of the standard expected improvement (EI) acquisition function for each of the $T$ objectives individually. Results in Figure B.9 demonstrate that `MOCOBO` performs significantly better with our proposed ECI acquisition function, demonstrating the importance of ECI to the performance of `MOCOBO`. Additionally, `MOCOBO-EIT` still performed fairly well, outperforming all other baseline methods considered in the paper (e.g., all baselines in Figure 2). This highlights the importance of other aspects MOCOBO.

### B.3 Execution Time of Greedy Approximation Algorithm

In Table B.3, we provide an empirical evaluation of the average wall-clock runtime of Algorithm 1 with different values of $N$ (the number of data points), $K$ (the covering set size), and $T$ (the number of objectives). We provide average run times with $N = 20, 200, 2000, 20000, 200000$, and $2000000$ for each combination of $K$ and $T$ that we ran `MOCOBO` on in Section 4. To improve the speed of Algorithm 1, in all experiments, we compute the inner for loop (lines 5-6 in Algorithm 1) in parallel. This involves simply computing the marginal coverage of adding all $N$ data points at once in parallel, and then updating $A$ with the point that achieved maximum marginal converge improvement. Parallel computation of the marginal coverage improvement of the $N$ points is straightforward to implement using PyTorch. In Table B.3, the "Runtime" column gives the average runtime of Algorithm 1 with this parallel computation of the of the inner for loop, and the "Runtime No Parallelization" column gives the average runtime without the parallel computation. Notice that as $N$ grows large, this parallel computation becomes essential to achieve the reasonably fast run times of Algorithm 1 needed to run `MOCOBO`. Average run times reported in Table B.3 are computed by running Algorithm 1 on a NVIDIA RTX A5000 GPU and averaging over $10$ runs.

Table B.3: Average wall-clock runtime of Algorithm 1 for different values of $N$ (the number of data points), $K$ (the covering set size), and $T$ (the number of objectives). The "Runtime" column gives the average runtime of Algorithm 1 with parallel computation of the of the marginal coverage of adding the $N$ data points (parallel computation of lines 5-6). The "Runtime No Parallelization" column gives the average runtime of Algorithm 1 without this parallelization of the inner for loop. Average run times are computed by running Algorithm 1 on a NVIDIA RTX A5000 GPU and averaging over 10 runs. Standard errors over the 10 runs are also provided.

| N | K | T | Runtime No Parallelization (seconds) | Runtime (seconds) |
|---|---|---|---|---|
| 20 | 2 | 4 | $0.00605 \pm 0.00385$ | $0.00864 \pm 0.00795$ |
| 200 | 2 | 4 | $0.0230 \pm 0.00391$ | $0.00901 \pm 0.00828$ |
| 2000 | 2 | 4 | $0.194 \pm 0.00389$ | $0.00850 \pm 0.00781$ |
| 20000 | 2 | 4 | $2.0523 \pm 0.0259$ | $0.00852 \pm 0.00780$ |
| 200000 | 2 | 4 | $20.845 \pm 0.0446$ | $0.00872 \pm 0.00796$ |
| 2000000 | 2 | 4 | $195.740 \pm 0.114$ | $0.00923 \pm 0.00792$ |
| 20 | 2 | 8 | $0.00596 \pm 0.00375$ | $0.00906 \pm 0.00833$ |
| 200 | 2 | 8 | $0.0269 \pm 0.00446$ | $0.00894 \pm 0.00808$ |
| 2000 | 2 | 8 | $0.197 \pm 0.00386$ | $0.00865 \pm 0.00794$ |
| 20000 | 2 | 8 | $1.947 \pm 0.00348$ | $0.00864 \pm 0.00789$ |
| 200000 | 2 | 8 | $19.508 \pm 0.00964$ | $0.00915 \pm 0.00825$ |
| 2000000 | 2 | 8 | $193.317 \pm 0.218$ | $0.00962 \pm 0.00785$ |
| 20 | 4 | 12 | $0.00861 \pm 0.00430$ | $0.00873 \pm 0.00785$ |
| 200 | 4 | 12 | $0.0440 \pm 0.00371$ | $0.00875 \pm 0.00789$ |
| 2000 | 4 | 12 | $0.386 \pm 0.00642$ | $0.00910 \pm 0.00821$ |
| 20000 | 4 | 12 | $4.303 \pm 0.120$ | $0.00869 \pm 0.00776$ |
| 200000 | 4 | 12 | $38.596 \pm 0.0895$ | $0.00889 \pm 0.00784$ |
| 2000000 | 4 | 12 | $395.325 \pm 0.169$ | $0.0111 \pm 0.00772$ |
| 20 | 3 | 6 | $0.00704 \pm 0.00384$ | $0.00887 \pm 0.00809$ |
| 200 | 3 | 6 | $0.0334 \pm 0.00372$ | $0.00840 \pm 0.00764$ |
| 2000 | 3 | 6 | $0.314 \pm 0.00654$ | $0.00897 \pm 0.00817$ |
| 20000 | 3 | 6 | $2.942 \pm 0.00478$ | $0.00872 \pm 0.00785$ |
| 200000 | 3 | 6 | $28.732 \pm 0.0156$ | $0.00862 \pm 0.00775$ |
| 2000000 | 3 | 6 | $312.825 \pm 0.203$ | $0.00985 \pm 0.00799$ |
| 20 | 4 | 7 | $0.00765 \pm 0.00370$ | $0.00880 \pm 0.00793$ |
| 200 | 4 | 7 | $0.0448 \pm 0.00365$ | $0.00847 \pm 0.00760$ |
| 2000 | 4 | 7 | $0.412 \pm 0.00252$ | $0.00887 \pm 0.00798$ |
| 20000 | 4 | 7 | $3.917 \pm 0.00461$ | $0.00882 \pm 0.00789$ |
| 200000 | 4 | 7 | $38.932 \pm 0.0402$ | $0.00889 \pm 0.00791$ |
| 2000000 | 4 | 7 | $389.272 \pm 0.588$ | $0.0105 \pm 0.00809$ |
| 20 | 4 | 11 | $0.00772 \pm 0.00377$ | $0.00955 \pm 0.00853$ |
| 200 | 4 | 11 | $0.0422 \pm 0.00383$ | $0.00859 \pm 0.00775$ |
| 2000 | 4 | 11 | $0.387 \pm 0.00382$ | $0.00931 \pm 0.00837$ |
| 20000 | 4 | 11 | $4.155 \pm 0.00758$ | $0.00887 \pm 0.00792$ |
| 200000 | 4 | 11 | $41.869 \pm 0.0203$ | $0.00880 \pm 0.00777$ |
| 2000000 | 4 | 11 | $381.313 \pm 0.199$ | $0.0114 \pm 0.00803$ |

**Efficiency of Algorithm 1** Algorithm 1 could also be made more efficient by pruning the points in $D_s$ with zero marginal coverage improvement after each outer-loop, since these points will continue to have zero marginal coverage improvement on subsequent loops. In future work, we plan to explore this and any other tricks that might allow us to further improve the efficiency of Algorithm 1.

## C   Batch Acquisition with Expected Coverage Improvement (ECI)

In batch acquisition, we select a batch of $q > 1$ candidates for evaluation. In Equation (7), we define q-ECI, a natural extension of the ECI acquisition function defined in Equation (5) to the batch acquisition setting. q-ECI gives the expected improvement in the coverage score after simultaneously observing the batch of $q$ points $\mathbf{X} = \{\mathbf{x}_1, \ldots, \mathbf{x}_q\}$. When using batch acquisition, we aim to select a batch of $q$ points that maximize q-ECI.

We will first discuss how one would estimate q-ECI using a Monte Carlo (MC) approximation. To select a batch of candidates $\hat{\mathbf{X}}$, we sample $m$ batches of $q$ points $B = \{\mathbf{B}_1, \mathbf{B}_2, ..., \mathbf{B}_m\}$. Here $\mathbf{B}_j$ is a batch of $q$ sampled points $\mathbf{B}_j = \{\mathbf{b}_{j1}, \ldots, \mathbf{b}_{jq}\}$. For each batch $\mathbf{B}_j$, we sample a realization $\hat{\mathbf{Y}}_j = \{\hat{\mathbf{y}}_{j1}, \ldots, \hat{\mathbf{y}}_{jq}\}$ from the GP surrogate model posterior. We leverage these samples to compute an MC approximation to the q-ECI of each $\mathbf{B}_j$:

$$q\text{-CI}(\mathbf{B}_j) = \max(0, \max_{r=1,\ldots,q} c(S^*_{D_s \cup \{(\mathbf{b}_{jr}, \hat{\mathbf{y}}_{jr})\}}) - c(S^*_{D_s})). \tag{8}$$

Here, $c(S^*_{D_s \cup \{(\mathbf{b}_{jr}, \hat{\mathbf{y}}_{jr})\}})$ is the approximation of the coverage score of the new best covering set if we choose to evaluate candidate $\mathbf{b}_{jr}$, assuming the candidate point will have the sampled objective values $\hat{\mathbf{y}}_{jr}$. We would like to select and evaluate the batch of candidates $\mathbf{B}_j$ with the largest q-CI.

Evaluating q-CI for a single candidate batch requires $q$ evaluations of $c(S^*_{D_s \cup \{(\mathbf{b}_{jr}, \hat{\mathbf{y}}_{jr})\}})$. Each evaluation of $c(S^*_{D_s \cup \{(\mathbf{b}_{jr}, \hat{\mathbf{y}}_{jr})\}})$ requires a call to Algorithm 1 to first construct $S^*_{D_s \cup \{(\mathbf{b}_{jr}, \hat{\mathbf{y}}_{jr})\}}$. Thus, batch acquisition with a full MC approximation of q-ECI requires $O(q \times m)$ calls of Algorithm 1. Assuming a sufficiently large $m$ to achieve a reliable MC approximation, this can become expensive for large batch sizes $q$. We therefore propose a faster approximation of batch ECI for practical use with large $q$.

Instead of sampling $m$ batches of candidates, we sample $m$ individual data points $P = \{\mathbf{p}_1, \mathbf{p}_2, ..., \mathbf{p}_m\}$. For each sampled point $\mathbf{p}_j$, we sample a realization $\hat{\mathbf{y}}_j = (\hat{f}_1(\mathbf{p}_j), \ldots, \hat{f}_T(\mathbf{p}_j))$. As in Section 3.2.1, we use the sampled realizations $\hat{\mathbf{y}}_j$ to compute an approximate coverage improvement $CI(\mathbf{p}_j)$ as defined in Equation (6) for each point $\mathbf{p}_j$. To obtain a batch of $q$ candidates, we then greedily select the $q$ points $\mathbf{p}_j \in P$ with the $q$ largest expected coverage improvements. Note that this strategy does not involve sequential optimization of the $q$ points, as the batch of $q$ points is selected simultaneously as the points with the top-$q$ expected coverage improvements.

## D  Limitations and Future Works

**Choosing $K$.** A primary limitation of `MOCOBO` is that the choice of the hyperparameter $K$ (the covering set size) may not always be straightforward. For example, as we mention in Section 4, it may require domain knowledge to choose $K < T$ that is large enough that achieving good coverage is possible despite multiple conflicting objectives. In many practical applications, we prefer the smallest possible set size $K$ such that good coverage can still be achieved. In future work, we plan to explore methods for simultaneously optimizing both the solutions in the covering set, and the size of the covering set, balancing the trade-off between minimizing the number of solutions needed and maximizing overall coverage.

**Unsupervised generative model pre-training for structured domains.** We note that applying `MOCOBO` to structured domains requires a pre-trained generative model, such as a Variational Autoencoder (VAE), to embed the discrete input space into a continuous latent space where `MOCOBO` can be directly applied. Training such generative models typically demands significant computational resources and a large corpus of unlabeled data, which may not always be available in new domains. For the two structured tasks considered in this paper—molecule and peptide design—we leverage publicly available pre-trained VAEs from prior work. In contrast, `MOCOBO` does not require a generative model for continuous input spaces, where optimization is performed directly in the original domain. This distinction highlights a key practical limitation: deploying `MOCOBO` in new structured domains would necessitate pre-training a new generative model, which may be a barrier in resource-constrained settings.

**Exploring lazy greedy evaluation approaches to improve computational efficiency.** Another avenue for future work is improving the computational efficiency of Algorithm 1 by using a lazy greedy evaluation strategy. Commonly employed in submodular maximization, lazy greedy approaches maintain a priority queue of marginal gains and only recompute them when necessary, avoiding redundant evaluations and reducing total computational cost.

**Threshold-based coverage.** The coverage optimization problem we consider in this paper provides a principled way to pose the goal of finding a small set of $K$ solutions such that each of the $T$ objectives is optimized *as much as possible* by at least one solution in the set. This captures the setting we care about, for example, discovering $K$ antibiotics such that each pathogen is targeted as effectively as possible, not just adequately. In contrast, threshold-based coverage constitutes a fundamentally different problem: it assumes that we know, a priori, a threshold value for each objective beyond which we do not care to improve performance further. This setting is not aligned with the domains we focus on in this paper where better objective values are always desirable and thresholds are typically unknown or unhelpful. However, threshold-based coverage is a meaningful formulation for many other domains, such as drug toxicity screening, where desired minimum performance thresholds are known ahead of time. To address such domains, we plan to explore this alternative problem setting of threshold-based coverage optimization in future work.

## E  Broader Impact

This research includes applications in molecule and peptide design. While AI-driven biological design holds great promise for benefiting society, it is crucial to acknowledge its dual-use potential. Specifically, AI techniques designed for drug discovery could be misused to create harmful biological agents [65].

Our goal is to accelerate drug development by identifying promising candidates, but it is imperative that experts maintain oversight, that all potential therapeutics undergo thorough testing and clinical trials, and that strict regulatory frameworks governing drug development and approval are followed.

# F  Compute Resources

In this section, we provide all details about the compute resources used to produce all results in this paper.

Table F.1: Setup of internal cluster used to run experiments.

| Type | Specifications |
|------|----------------|
| System Topology | 20 nodes with 2 sockets each with 24 logical threads (total 48 threads) |
| Processor | 1 Intel Xeon Silver 4310, 2.1 GHz (maximum 3.3 GHz) per socket |
| Cache | 1.1 MiB L1, 30 MiB L2, and 36 MiB L3 |
| Memory | 250 GiB RAM |
| Accelerator | 1 NVIDIA RTX A5000 per node, 2 GHZ, 24GB RAM |

**Compute specifications (type and memory).**  We use GPU works to run all experiments and produce all empirical results provided in this paper. A single GPU was used per run of each method compared on each task. Each each run uses approximately 12-18 GB of the GPU memory. Most experiments were executed on our internal cluster of NVIDIA RTX A5000 GPUs (see internal cluster compute details in Table F.1). We also used cloud compute resources for two weeks to complete additional replications of some experiments. We used a total of eight RTX 4090 GPU workers from `runpod.io`, each with approximately 24 GB of GPU memory.

**Execution time.**  For the relatively inexpensive rover task, each optimization run takes approximately 1 day of execution time. For all other tasks considered, each optimization run takes approximately 3 days of execution time. To create all coverage optimization plots, we ran all methods compared 20 times each. Completing all of the runs needed to produce all of the results in this paper required roughly 64000 total GPU hours.

**Compute resources for preliminary experiments.**  Preliminary experiments refer to the initial experiments for e.g. method development that are not included as results in the paper. All preliminary experiments were run on our internal cluster of NVIDIA RTX A5000 GPUs (see internal cluster compute details in Table F.1). We spent approximately 2000 hours of GPU time on preliminary experiments.

# G   Additional Implementation Details

In this section, we provide additional implementation details for `MOCOBO`. We also refer readers to the `MOCOBO` codebase for the full-extent of implementation details and experimental setup needed to reproduce results provided `https://github.com/nataliemaus/mocobo`.

## G.1   Trust Region Hyperparameters

For all trust region methods, the trust region hyperparameters are set to the `TuRBO` defaults used by Eriksson et al. [21].

## G.2   Surrogate Model

Since the tasks considered in this paper are challenging, high-dimensional tasks requiring a large number of function evaluations, we use approximate Gaussian process (GP) surrogate models. In particular, we use Parametric Gaussian Process Regressor (PPGPR) [35] surrogate models with a constant mean, standard RBF kernel, and $1024$ inducing points. Additionally, we use a deep kernel (several fully connected layers between the search space and the GP kernel) [66]. We use two fully connected layers with $D$ nodes each, where $D$ is the dimensionality of the search space.

We use the same PPGPR model(s) with the same configuration for `MOCOBO`, `TuRBO`, `LOL-BO`, and `ROBOT`. For `MOCOBO`, to model the $T$-dimensional output space, we use $T$ PPGPR models, one to approximate each objective $f_1, \ldots, f_T$. To allow information sharing between the models, we use a shared deep kernel (the $T$ PPGPR models share the same two-layer deep kernel) [67, 68].

Unlike the other methods compared, `MORBO` was designed for use with an exact GP model rather than an approximate GP surrogate model. For fair comparison, we therefore run `MORBO` with an exact GP using all default hyperparameters and the official codebase provided by Daulton et al. [14].

We train the PPGPR surrogate model(s) on data collected during optimization using the Adam optimizer [69] with a learning rate of $0.001$ and a mini-batch size of $256$. On each step of optimization, we update the model on collected data until we stop making progress (loss stops decreasing for $3$ consecutive epochs), or exceed $30$ epochs. Since we collect a large amount of data for each optimization run (e.g., as many as $2e6$ data points in a single run for the "template constrained" peptide design task), we avoid updating the model on all data collected at once. On each step of optimization, we update the current surrogate model only on a subset of $1000$ of the collected data points. This subset is constructed from the data that has obtained the highest objective values so far, along with the most recent batch of data collected. By always updating on the most recent batch of data collected, we ensure that the surrogate model is conditioned on every data point collected at some point during the optimization run.

## G.3   Initialization Data

In this section, we provide details regarding the data used to initialize all optimization runs for all tasks in Section 4.

To initialize optimization for the molecule design task, we take a random subset of $10000$ molecules from the standardized unlabeled dataset of 1.27M molecules from the Guacamol benchmark software [52]. We generate labels for these $10000$ molecules once, and then use the labeled data to initialize optimization for all methods compared.

To initialize optimization for the peptide design tasks, we generate a a set of $20000$ peptide sequences by making random edits (insertions, deletions, and mutations) to the $10$ template peptide sequences in Table H.4. We generate labels for these $20000$ peptides once, and then use the labeled data to initialize optimization for all methods compared.

For all other tasks, we initialize optimization with $2000$ points sampled uniformly at random from the search space.

### G.4 Diversity Constraints and Associated Hyperparameters for the `ROBOT` Baseline

For a single objective, `ROBOT` seeks a diverse set of $M$ solutions, requiring that the set of solutions have a minimum pairwise diversity $\tau$ according to the user specified diversity function $\delta$. Since Maus et al. [28] also consider rover and molecule design tasks, we use the same diversity function $\delta$ and diversity threshold $\tau$ used by Maus et al. [28] for these two tasks. For the peptide design tasks, we define $\delta$ to be the edit distance between peptide sequences, and use a diversity threshold of $\tau = 3$ edits. For the image optimization task, since there is no obvious semantically meaningful measure of diversity between two sets of input parameters, we define $\delta$ to be the Euclidean distance between solutions, and use $\tau = 1.45$, the approximate average Euclidean distance between a randomly selected pair of points in the search space.

# H  Additional Task Details

In this section we provide additional details for the chosen set of tasks we provide results for in Section 4.

## H.1  HDR Image Tone Mapping

Table H.1: `pyiqa` metric ID strings used to identify the $T = 7$ target image quality metrics used for image tone mapping. Each metric is a no-reference image aesthetic (IAA) or quality (IQA) assessment metric obtained from the `pyiqa` library [55].

| Objective ID | Pyiqa Metric ID | Reference |
|---:|---|---|
| 1 | `nima` | 70 |
| 2 | `nima-vgg16-ava` | 70, 71 |
| 3 | `topiq-iaa-res50` | 72 |
| 4 | `laion-aes` | 73 |
| 5 | `hyperiqa` | 74 |
| 6 | `tres` | 75 |
| 7 | `liqe` | 76 |

In Table H.1, we list the names of the 7 image quality metrics used for the image tone mapping tasks described in Section 4.1.

**Target metrics.**   The target image quality (Image Quality Assessment, IQA) and aesthetic (Image Aesthetic Assessment, IAA) metrics are organized in Table H.1, where all except `nima` are IAA metrics, while `nima` is an IQA metric. For more detailed information about each metric and their corresponding datasets, please refer to their original references.

**Benchmark images.**   We use two benchmark images. The first is the "Stanford Memorial Church" image obtained from `https://www.pauldebevec.com/Research/HDR/` by courtesy of Paul E. Debevec [56]. The second is the "desk lamp" image obtained from `https://cadik.posvete.cz/tmo/` by courtesy of Martin Čadík [57]. Because commonly used metrics are only correlated with subjective image quality, prior work on tuning parameters for these and related benchmarks has been done by trial and error and human-in-the-loop type schemes [77], including preferential BO-based approaches [78, 79].

**Imaging pipeline.**   We consider a tone mapping pipeline consisting of a multi-layer detail decomposition [80–82] using the guided filter by He et al. [83] (3 detail layers and 1 base layer), followed by gamma correction [54, Section 2.9], resulting in a 13-dimensional optimization problem. The complete image processing pipeline is very similar to the classic approach proposed by Tumblin and Turk [80], where the main difference is that, similarly to Farbman et al. [84], we replace the diffusion smoothing filter with a more recent edge-preserving detail smoothing filter, the guided filter, by He et al. [83]. First, given an HDR image in the RGB color space $I = (I_r, I_r, I_b))$, where $I_r$ is the red channel, $I_r$ is the blue channel, and $I_g$ is the green channel, we compute the luminance according to

$$L \triangleq 0.2989 I_r + 0.587 I_g + 0.114 I_b \,.$$

(The constants were taken from the code of Farbman et al. 84.) The luminance image $L$ is then logarithmically compressed and then decomposed into three detail layers and one base layer by applying the guided filter three times, each with a different radius parameter $r_i$ and smoothing parameter $\epsilon_i$ for $i = 1, \ldots, 3$. Then, we amplify or attenuate each channel with a corresponding gain coefficient $g_{\text{detail},i}$ $i = 1, \ldots, 3$ for the detail and $g_{\text{base}}$ for the base layers. The image is then reconstructed by adding all the layers, including the colors. Following Tumblin and Turk [80], we also apply a gain, $g_{\text{color}}$, to the color channels. Finally, the resulting image is applied an overall gain $g_{\text{out}}$ and then gamma-corrected [54, Section 2.9] with an exponent of $1/\gamma$. The parameters for this pipeline are organized in Table H.2. The implementation uses OpenCV [85], in particular, the guided filter implementation in the extended image processing (`ximgproc`) submodule.

Table H.2: `pyiqa` metric ID strings used to identify the $T = 7$ target image quality metrics used for the image tone mapping task.

| Parameter | Description | Domain |
|---|---|---|
| $r_i$ | radius of the guided filter for generating the $i$th detailed layer | $\{3, \ldots, 32\}$ |
| $\epsilon_i$ | $\epsilon$ of the guided filter for generating the $i$th detailed layer | $[0.01, 10]$ |
| $g_{\text{detail},i}$ | Gain of the $i$th detail layer | $[0, 1.5]$ |
| $g_{\text{base}}$ | Gain of the $i$th detail layer | $[0, 1]$ |
| $g_{\text{color}}$ | Gain of the color layer | $[0.5, 1.5]$ |
| $g_{\text{out}}$ | Gain of tone-mapped output | $[0.2, 2.0]$ |
| $\gamma$ | Gamma correction inverse exponent | $[1, 5]$ |

**Optimization problem setup.** For optimization, we map the parameters

$$\mathbf{x} = (r_1, \, r_2, \, r_3, \, \epsilon_1, \, \epsilon_2, \, \epsilon_3, \, g_{\text{detail},1}, \, g_{\text{detail},2}, \, g_{\text{detail},3}, \, g_{\text{base}}, \, g_{\text{color}}, \, g_{\text{out}}, \, \gamma)$$

to the unit hypercube $[0, 1]^{13}$. In particular, the mapped values on the unit interval $[0, 1]$ linearly interpolate the domain of each parameter shown in Table H.2. For the radius parameters $r_1, r_2, r_3$, which are categorical, we naively quantize the domain by rounding the output of the interpolation to the nearest integer.

## H.2 Peptide Design

Table H.3: Names of the $T = 11$ target bacteria used for the peptide design task. The first seven bacteria are Gram negative (IDs B1-B7) and the last four (IDs B8-B11) are Gram positive.

| Objective ID | Target Pathogenic Bacteria |
|---|---|
| B1 | A. baumannii ATCC 19606 |
| B2 | E. coli ATCC 11775 |
| B3 | E. coli AIC221 |
| B4 | E. coli AIC222-CRE |
| B5 | K. pneumoniae ATCC 13883 |
| B6 | P. aeruginosa PAO1 |
| B7 | P. aeruginosa PA14 |
| B8 | S. aureus ATCC 12600 |
| B9 | S. aureus ATCC BAA-1556-MRSA |
| B10 | E. faecalis ATCC 700802-VRE |
| B11 | E. faecium ATCC 700221-VRE |

Table H.4: Template amino acid sequences used for the "template constrained" peptide design task.

| Template Amino Acid Sequences |
|---|
| RACLHARSIARLHKRWRPVHQGLGLK |
| KTLKIIRLLF |
| KRKRGLKLATALSLNNKF |
| KIYKKLSTPPFTLNIRTLPKVKFPK |
| RMARNLVRYVQGLKKKKVI |
| RNLVRYVQGLKKKKVIVIPVGIGPHANIK |
| CVLLFSQLPAVKARGTKHRIKWNRK |
| GHLLIHLIGKATLAL |
| RQKNHGIHFRVLAKALR |
| HWITINTIKLSISLKI |

Table H.3 specifies the $T = 11$ target bacteria used for the peptide design task from Section 4. The first seven bacteria are Gram negative bacteria (Objective IDs B1-B7) and the last four (Objective IDs B8-B11) are Gram positive. Table H.4 gives the 10 template amino acid sequences used for the "template constrained" variation of the peptide design task from Section 4.

# I  Proof that Finding the Best Observed Covering Set is NP-hard

In this section, we prove Theorem 3.1, that finding $S^*_{D_s}$ is NP-Hard.

*Proof.* We prove Theorem 3.1 by reduction from the well-known *Maximum Coverage Problem (MCP)*, which is NP-Hard.

**Definition I.1** (Maximum Coverage Problem (MCP))**.** In the Maximum Coverage Problem, we are given:

- A universe $U = \{e_1, e_2, \ldots, e_m\}$ of $m$ elements.

- A collection of $n$ subsets $\mathcal{S} = \{A_1, A_2, \ldots, A_n\}$, where $A_i \subseteq U$.

- An integer $K$, the number of subsets we can select.

The objective is to find a collection of $K$ subsets $\mathcal{S}' \subseteq \mathcal{S}$ such that the total number of elements covered, $\bigcup_{A \in \mathcal{S}'} A$, is maximized.

**Proposition I.2** (MCP is NP-Hard)**.** *The Maximum Coverage Problem is NP-Hard.*

**Reduction from MCP to Finding $S^*_{D_s}$:**  We reduce an instance of MCP to the problem of finding $S^*_{D_s}$ as follows:

1. Let the universe $U = \{e_1, e_2, \ldots, e_m\}$ correspond to the objectives $\{1, 2, \ldots, T\}$ in the optimal covering set problem, i.e., set $T = m$.

2. Let each subset $A_i \in \mathcal{S}$ correspond to a point $\mathbf{x}_i \in D_s$.

3. Define each objective $f_t : \mathcal{X} \to \{0, 1\}$, and set $f_t(\mathbf{x}_i) = 1$ if subset $A_i$ contains element $e_t$, and $f_t(\mathbf{x}_i) = 0$ otherwise. Intuitively, this means that each point $\mathbf{x}_i$ "covers" objective $f_t$ if it achieves value 1 under that objective. While the design points $x_i$ are not literal subsets, they induce coverage behavior that mirrors the structure of MCP through their binary function values across the objectives.

4. Under this mapping, coverage score $c(S) = \sum_{t=1}^{T} \max_{\mathbf{x} \in S} f_t(\mathbf{x})$ is the total number of objectives "covered" (i.e., the number of objectives for which at least one of the selected points has value 1). It follows that the goal of the Maximum Coverage Problem (selecting $K$ "subsets" to maximize coverage) corresponds exactly to selecting $K$ points $\mathbf{x}_i \in D_s$ to maximize the coverage score $c(S)$.

**Correctness of the Reduction:**  The reduction ensures that:

- Each subset $A_i \in \mathcal{S}$ is encoded by a design point $\mathbf{x}_i \in D_s$ via its binary-valued outputs over the objectives.

- Each element $e_t \in U$ is mapped to objective $f_t$, and is considered "covered" if some selected point $\mathbf{x} \in S$ satisfies $f_t(\mathbf{x}) = 1$. In particular, if subset $A_i$ covers/contains element $e_t$, then $f_t(\mathbf{x}_i) = 1$; otherwise, $f_t(\mathbf{x}_i) = 0$.

- The MCP objective (maximize number of covered elements) is equivalent to maximizing the coverage score $c(S)$, which counts how many objectives are covered by the selected set $S$.

Thus, solving the optimal covering set problem is equivalent to solving MCP.

**Implications:**  Since MCP is NP-Hard (Proposition I.2), and we have reduced MCP to the problem of finding $S^*_{D_s}$ in polynomial time, it follows that finding $S^*_{D_s}$ is also NP-Hard.

$\square$

**Proof significance** We do not claim the above proof of Theorem 3.1 as a novel contribution of this work, as it follows straightforwardly from the fact that the well-known Maximum Coverage Problem is NP-hard. Note that proving Theorem 3.1 would also follow straightforwardly from the work of Ding et al. [17] who demonstrate that the coverage optimization problem generalizes k-means clustering. Rather than providing additional novel contribution, the above proof of Theorem 3.1 serves to justify our use of a greedy approximation algorithm (Algorithm 1) to approximate $S_{D_s}^*$ on each iteration of MOCOBO.

# J   Approximation Proof for Greedy Algorithm

In this section, we prove Theorem 3.2, that Algorithm 1 is a $(1 - \frac{1}{e})$-Approximation.

**Definition J.1** (Coverage Score). The coverage score of a set $S \subseteq D_s$, denoted $c(S)$, is defined as:

$$c(S) = \sum_{t=1}^{T} \max_{\mathbf{x} \in S} f_t(\mathbf{x}),$$

where $f_t(\mathbf{x})$ is the observed value of objective $t$ at point $\mathbf{x}$.

**Definition J.2** (Optimal Covering Set). Let $S_{D_s}^* \subseteq D_s$ denote the optimal covering set of size $K$:

$$S_{D_s}^* = \arg\max_{S \subseteq D_s, |S|=K} c(S).$$

Its coverage score is given by $c(S_{D_s}^*)$.

**Definition J.3** (Contribution of Objective Function $f_t$ to Coverage Score). Given a covering set $S \subseteq D_s$, we denote the contribution of objective $f_t \in \{f_1, f_2, \dots, f_T\}$ to the overall coverage score $c(S)$ as $g_t(S)$ where

$$g_t(S) = \max_{\mathbf{x} \in S} f_t(\mathbf{x}).$$

It follows that from Theorem J.3 that:

$$c(S) = \sum_{t=1}^{T} g_t(S).$$

**Lemma J.4** (Monotonicity). *The coverage score $c(S)$ as defined in Theorem J.1 is monotone, i.e., for any $S \subseteq S' \subseteq D_s$, we have:*

$$c(S) \leq c(S').$$

*Proof.* Adding more points to a set can only increase or maintain the maximum values of $f_t$ for each objective $t$, since for each objective:

$$g_t(S') = \max\left(g_t(S), g_t(S' \setminus S)\right) \geq g_t(S).$$

Hence, $c(S)$ is monotone. $\qquad\square$

**Lemma J.5** (Submodularity). *The coverage score $c(S)$ as defined in Theorem J.1 is submodular, i.e., for any $S \subseteq S' \subseteq D_s$ and any $\mathbf{x}^* \in D_s \setminus S'$, we have:*

$$c(S \cup \{\mathbf{x}^*\}) - c(S) \geq c(S' \cup \{\mathbf{x}^*\}) - c(S').$$

*Proof.* The marginal improvement of adding $\mathbf{x}^*$ to $S$ is:

$$
\begin{aligned}
c(S \cup \{x^*\}) - c(S) &= \sum_{t=1}^{T} \left(g_t(S \cup \{x^*\}) - g_t(S)\right) \\
&= \sum_{t=1}^{T} \max\left[f_t(x^*), g_t(S)\right] - g_t(S) \\
&= \sum_{t=1}^{T} \begin{cases} f_t(\mathbf{x}^*) - g_t(S) & f_t(\mathbf{x}^*) > g_t(S) \\ 0 & \text{otherwise} \end{cases}.
\end{aligned}
$$

Likewise, for $S'$ this is:

$$c(S' \cup \{x^*\}) - c(S') = \sum_{t=1}^{T} \begin{cases} f_t(\mathbf{x}^*) - g_t(S') & f_t(\mathbf{x}^*) > g_t(S') \\ 0 & \text{otherwise} \end{cases}.$$

Now, noting that for each term

$$g_t(S') = \max\left[g_t(S), g_t(S' \setminus S)\right] \geq g_t(S),$$

we know then that $f_t(\mathbf{x}^*) - g_t(S') \le f_t(\mathbf{x}^*) - g_t(S)$, and further that $f_t(\mathbf{x}^*) > g_t(S')$ implies that $f_t(\mathbf{x}^*) > g_t(S)$. Taken together, these imply that:

$$\sum_{t=1}^{T} \begin{cases} f_t(\mathbf{x}^*) - g_t(S') & f_t(\mathbf{x}^*) > g_t(S') \\ 0 & \text{otherwise} \end{cases} \le \sum_{t=1}^{T} \begin{cases} f_t(\mathbf{x}^*) - g_t(S) & f_t(\mathbf{x}^*) > g_t(S) \\ 0 & \text{otherwise} \end{cases}$$

and therefore:

$$c(S \cup \{\mathbf{x}^*\}) - c(S) \ge c(S' \cup \{\mathbf{x}^*\}) - c(S').$$

$\square$

**Lemma J.6** (Greedy Achieves a $(1 - \frac{1}{e})$-Approximation for Monotone Submodular Functions). *For any monotone submodular function, the greedy submodular optimization strategy provides a $(1 - \frac{1}{e})$-approximation.*

*Proof.* This is a well-known result about monotone submodular functions shown, for example, by Nemhauser et al. [27]. $\square$

*Proof.* **Proof of Theorem 3.2** From Theorem J.4 and Theorem J.4, we know that $c(S)$ is monotone submodular.

Since $c(S)$ is monotone submodular, and Algorithm 1 approximates $A^*_{D_s}$ using greedy submodular optimization, it follows from Theorem J.6 that Algorithm 1 achieves:

$$c(A^*_{D_s}) \ge \left(1 - \frac{1}{e}\right) c(S^*_{D_s}).$$

$\square$

