# OpenReview forum: "Covering Multiple Objectives with a Small Set of Solutions Using Bayesian Optimization"
_NeurIPS.cc/2025/Conference — NeurIPS 2025 poster_

### Official Review · Reviewer_9SUK · 2025-06-25

**Clarity:** 4
**Significance:** 4
**Originality:** 3
**Rating:** 6
**Confidence:** 4

**Summary:**

This paper proposes a BO-inspired algorithm to do multi-objective covering optimization. The authors define a "coverage" success metric

$$c({x_1, \ldots, x_K}) = \sum_{t=1}^T\max_{k=1}^K f_t(x_k)$$

and their proposed algorithm is essentially expected improvement of coverage, estimating the expectation using Monte Carlo and maximizing over subsets using a submodular greedy approach which should achieve $(1-1/e)$ accuracy.

The authors perform a variety of experiments, _including one with real in-vitro results_!

**Questions:**

1: batch extension

This is described in the text as ECI after simultaneously observing the batch. However, shouldn't that quantity be $c(S^*_{D_s \cup_{i=1}^q(x_i, y_i)})$ instead of what is in equation 6? (a maximum over elements in the batch). The objective in equation 6 seems to be "ECI if you simultaneously observe labels and are then allowed to choose 1 element to keep"

2: Redundancy in ECI equation

In equation 4 you write:

$$\max(0, c(S_{D \cup (x,y)}) - c(S_{D}))$$

Because $c$ is submodular, doesn't that imply that this quantity is always greater than 0? Therefore the max is not necessary. Since $c(S_{D})$ does not depend on $y$ it can also be moved outside the expectation. This would reduce the expression to just "expected coverage score" - a constant. Does that make sense? And does that suggest that the algorithm might under-explore?

3: Single-objective baselines

I don't understand how the comparison to TurBO / other single-objective BO methods was done. You stated that $T$ separate rounds of optimization were performed. How is sample efficiency measured for plotting in figure 2? Are you assuming a meta-algorithm where which cycles through taking one step for each objective?

**Ethical Concerns:**

["NO or VERY MINOR ethics concerns only"]

**Final Justification:**

My initial assessment was positive (highest possible score), and comments from the other reviewers have not convinced me to lower my score.

**Limitations:**

I think the limitations are adequately mentioned and addressed in the paper.

**Paper Formatting Concerns:**

- SI units are formatted a bit weirdly, especially in the appendix (eg $\mu mol = \mu \times m \times o \times l$ ). I suggest you use the [siunitx](https://ctan.org/pkg/siunitx) package.
- Everything is in one appendix subsection (A). Consider promoting all headings one level? Eg A.2 -> appendix B, A.2.1 -> B.1, etc. Just change all `\subsection` commands to `\section` and `\subsubsection` to `\subsection`.

**Quality:**

4

**Strengths And Weaknesses:**

Overall this is a really strong paper.

- Paper is very well-written and well-presented
- Algorithm proposed is novel, interesting, and technically sensible
- Good theoretical justification for greedy approximation
- Extensive experiments on a variety of tasks in different domains, including actual in-vitro lab experiments.

The only "weaknesses" I can see are fairly minor:

- I'm not super convinced about the practicality of this problem in drug discovery: the broad-spectrum antibiotic example makes sense in principle, but medicines are not usually developed _jointly_ like this, instead each program usually wants _one_ solution for _one_ objective. I don't see this being applied in a real-world company for that reason. That's nothing against the algorithm itself though.
- As with anything in BO, experimental comparisons are overshadowed by dependence on a large number of obscure hyperparameters and may not generalize, so no guarantees that this difference will translate to other settings. I'm not holding that against this paper too much because that is a more widespread issue in the field.
- Batch extension does not make a ton of sense (see question below)

---

> ### Author Rebuttal · Authors · 2025-07-30
>
> We sincerely thank the reviewer for the thoughtful feedback. We especially appreciate your recognition of the real-world *in-vitro* results. Below, we respond to each of your questions:
> # Q1: Batch Extension and Equation 6
> You’re absolutely right to flag this subtlety. The form in Equation 6 approximates the marginal gain in coverage under the simplifying assumption that only the best point in the batch contributes to coverage, rather than the full set. This is computationally much simpler to optimize in practice, but as you point out, it does not fully reflect the potential joint contribution of all q points. Your suggested form would indeed capture the “true” expected joint gain more faithfully. We chose the current formulation as a practical approximation. That said, exploring more accurate batch ECI objectives, like the one you suggest, is a very interesting direction for future work.
> # Q2: Redundancy in the ECI Expression
> Thank you for this insightful question. You’re correct that, in theory, the “max(0, …)” term in the ECI expression may not be strictly necessary, since the coverage score can only be increased or remain the same after a new data point is added. Similarly, since the baseline coverage value (before adding the new point) is fixed, one could factor it out of the expectation and simplify the expression.
> We kept the original form primarily for consistency with standard Expected Improvement-style notation and to make the marginal gain in coverage explicit. In practice, the predictive distribution over objective values still introduces enough variability that the expectation continues to provide a meaningful acquisition signal, so we haven’t observed any under-exploration behavior due to this formulation. That said, your observation is well-taken, and this is something we plan to explore further in future work.
> # Q3: Single-Objective Baselines and Figure 2
> Thank you for asking for clarification here. For each baseline (e.g., TuRBO), we perform T independent runs, one per objective, each with its own budget. To ensure a fair comparison on sample efficiency, we combine the results from these T runs by selecting the best K solutions (among the T collected ones) that maximize coverage. We then report the cumulative number of queries across all objectives (i.e., total function evaluations across the T runs) on the x-axis of Figure 2. So, there is no meta-algorithm cycling through objectives in a round-robin fashion. Instead, we assume access to T independent runs and aggregate the solutions post hoc to compute coverage, essentially giving the baselines every possible advantage while still showing that MOCOBO outperforms them.
> # Formatting and SI Units
> Thank you for pointing out the SI unit formatting issues and appendix structure. We’ll take your suggestions into account when preparing the camera-ready version, including exploring the use of the siunitx package and reorganizing the appendix for improved readability.

---

### Official Review · Reviewer_SYMT · 2025-07-01

**Clarity:** 2
**Significance:** 2
**Originality:** 2
**Rating:** 3
**Confidence:** 4

**Summary:**

The paper proposes MOCOBO, a Bayesian optimization (BO) method for coverage optimization. MOCOBO finds a small set of solutions where each objective is well-addressed by at least one solution. The work critiques traditional Pareto-front approaches for failing in extreme trade-off scenarios (e.g., drug discovery) and combines a novel Expected Coverage Improvement (ECI) acquisition function with trust-region BO. Experiments show MOCOBO matches or nears the performance of individually optimizing all objectives, outperforming adapted baselines.

**Questions:**

See also the weaknesses section above.
- Can the authors validate ECI's contribution compared to standard EI or entropy-based methods? Does it provide better acquisition behavior than those standard alternatives?
- Can the authors comment on the wall-clock time of the greedy cover construction vs. baselines such as MORBO or TurBO?
- CluSO is the true competitor of MOCOBO, as the other methods (TuRBO, MORBO, ROBOT) are not specifically designed for coverage, making the comparison unfair. However, more emphasis on comparison against CluSO could have been provided. Why does CluSO underperform (poor scalability, bad hyperparameters)?
- Are the trust regions necessary for MOCOBO? Or is MOCOBO just TuRBO with a coverage objective?

**Ethical Concerns:**

["NO or VERY MINOR ethics concerns only"]

**Final Justification:**

I finalized my score to weak reject because (i) I still have unclarified concerns, and (ii) I think that the current version of the manuscript requires "major revision" considering NeurIPS standards. Namely, the sensitivity of the method to batch size and initialization dataset size is left as future work. Importantly, the robustness of MOCOBO to the choice of K, which is an essential decision for the proposed approach, is not clarified to me. There is no regret-bound discussion that would help the theoretical understanding of the method. Instead of providing a small discussion, the authors rebut, stating that "recent popular works do not include bounds", which is not an effective way of clarifying a concern. Furthermore, as also raised by other reviewers, in terms of the writing, the problem definition and motivation require major revision to improve the clarity.

**Limitations:**

Yes, limitations, mainly the design choice of K, are discussed in Appendix A4. However, an additional discussion on computational scalability and domain-specific tuning (e.g., VAEs in structured domains) could have been placed.

**Paper Formatting Concerns:**

Table captions should be above the tables. It is not consistent with the NeurIPS paper style.

**Quality:**

2

**Strengths And Weaknesses:**

Strengths:
- The paper addresses a practically motivated problem which has strong relevance in domains with hard-to-satisfy trade-offs (e.g., pathogen-specific drug development).
- Experiments are conducted on a diverse set of tasks, considering different baselines.
- Empirical results show that MOCOBO consistently achieves near-optimal coverage, often matching the performance of T individually optimized solutions, compared to baselines.

Weaknesses:
- There is a lack of justification for some key design choices.
- - A justification on the design of the coverage score could have been provided. Specifically, no ablation is done over this key design. Alternative coverage metrics (e.g., threshold-based coverage) would have revealed whether MOCOBO's gains depend heavily on this definition, also given the fact that some of the baselines (e.g., MORBO) are not directly designed for the coverage problem but adapted for empirical evaluation.
- - Ablation comparing MOCOBO with/without trust regions to isolate the impact of trust regions.

- There are some missing ablations which have strengthened the analysis of the proposed method.
- - Batch size (q) ablation: How would the performance of the MOCOBO change compared to baselines, under various batch sizes? How do the performance and runtime scale with q?
- -  K ablation: It’s unclear how robust MOCOBO is to the choice of K, or how performance degrades when K is poorly chosen. The discussion on the ablation of K is provided in Section 4.3. However, since it is essential to the proposed approach, I think it would have been supported significantly better if experiments under different K values for a specific problem domain had been provided in the main text.
- -  Effectiveness of ECI against alternative acquisition functions could have supported the benefit of the proposed approach better. Ablating the proposed ECI acquisition against, e.g. maximum of standard EI per objective acquisition function, would clarify how much ECI contributes to performance vs. the trust-region or greedy cover construction.
- -  In structured tasks, the results heavily depend on VAE quality. Ablating the surrogate model quality (e.g., varying initial data size) would have clarified the benefits of MOCOBO from the embedding quality.

-  The problem domains cover continuous and structured search spaces. However, to clarify the generalizability of the proposed approach, experiments on a problem domain involving a mixed search space could have been provided.

- I think the novelty is overly stated. As provided with related works, similar ideas exist in clustering, gradient-based multi-objective optimization, and CluSO, which is proposed for black-box coverage.
- - Algorithm 1 is standard, specifically, a greedy algorithm for submodular maximization is textbook material (Nemhauser et al. 1978).
- - Instead, clearly motivating the idea with extreme objective conflict settings would highlight the contribution.

- No regret bounds or convergence guarantees are provided, unlike classic BO.

Minor:
- Typo in line 23, should be corrected as "optimization".
- In terms of writing, it is a bit repetitive. Particularly, the coverage problem is redefined excessively.

> Nemhauser, G.L., Wolsey, L.A. & Fisher, M.L. An analysis of approximations for maximizing submodular set functions—I. Mathematical Programming 14, 265–294 (1978).

---

> ### Author Rebuttal · Authors · 2025-07-30
>
> We thank the reviewer for their thoughtful feedback and for highlighting the practical relevance and strong empirical performance of MOCOBO. We respond to each point below:
> # 1. Coverage Score
> We appreciate the reviewer’s suggestion to examine alternative definitions of coverage. However, we respectfully clarify that our coverage score is **not a design choice**, but rather the **problem definition itself**, grounded in prior work. Specifically, the continuous coverage score we use was originally formalized in the CluSO paper and is a principled way to pose the goal of finding a small set of K solutions such that **each of the T objectives is optimized as much as possible** by at least one solution in the set. This captures the setting we care about, for example, discovering K antibiotics such that each pathogen is targeted as effectively as possible, not just adequately. By contrast, threshold-based coverage is indeed a meaningful formulation for some domains but it constitutes a fundamentally different problem: it assumes that we know, a priori, a threshold value for each objective beyond which we do not care to improve performance further. This makes sense in some safety-critical settings (e.g., drug toxicity screening), but is not aligned with the domains we focus on where **better objective values are always desirable** and thresholds are typically unknown or unhelpful. For these reasons, we adopted the coverage formulation from CluSO and did not include threshold-based comparisons. That said, we appreciate this point and now explicitly clarify in the paper that the problem definition we adopt is the same as has been used in the black-box coverage literature (CluSO), and we better motivate its relevance to our domains of interest. We also agree with the reviewer that methods like MORBO are not designed for coverage, and we will further emphasize in the main text that these are adapted baselines used to highlight that off-the-shelf BO methods do not trivially solve the coverage problem. We include them not to criticize, but to underscore the importance of explicitly addressing this problem class. We do directly compare to CluSO, which is the only existing method we are aware of that does try to solve the black-box coverage optimization problem directly.
> # 2. Trust Regions (TRs)
> Thank you for raising this point. We have added an ablation study comparing MOCOBO with and without trust regions (TRs) on the rover task with 4 obstacle courses, and on the template free peptide design task. As expected, MOCOBO performs substantially better with TRs, confirming that TRs significantly improve performance in the high-dimensional settings we consider. We have added plots and discussion of this to the appendix. We report below the average coverage score achieved by MOCOBO with and without TRs after N function evaluations (higher is better).
> ### Rover Task
> * N=40K, with TRs: $17.52  \pm 0.32$, without TRs: $-6.95  \pm 0.92$.
> * N=80K, with TRs: $17.76  \pm 0.31$, without TRs: $-4.64  \pm 0.69$.
> ### Peptide Task
> * N=300K, with TRs: $-45.30  \pm 6.38$, without TRs: $-242.53  \pm 11.56$.
> * N=600K, with TRs: $-39.50  \pm 5.74$, without TRs: $-225.84  \pm 13.30$.
>
> To clarify the motivation for using TRs: it is well-established in the BO literature that standard BO (without TRs or other high-dim adaptations) performs poorly in high dims. [1] originally introduced TRs as a principled way to improve performance of high-dim BO. Since then, TRs have become a standard tool for any high-dim BO task. Since all of the tasks we consider are high-dimensional, this precisely why we adopt TRs. We note that recent work has proposed alternative approaches to improve high-dim BO without TRs [2]. However, there remain concerns about the applicability of these approaches to structured domains (e.g., see Table 2 in [3], results show that [2] performs surprisingly poorly on structured tasks).
> # 3. Additional Ablations
> We added an ablation comparing to “MOCOBO-EIT": MOCOBO with the reviewer’s suggested "maximum of standard EI per objective" acquisition function. We ran this ablation on the rover task with 4 obstacle courses and on the template free peptide design task. We have added plots and discussion of this to the appendix. We report below the average coverage score achieved by MOCOBO (as proposed with ECI), and by MOCOBO-EIT, after N function evaluations (higher is better).
> ### Rover Task
> * N=40K, MOCOBO: $17.52  \pm 0.32$, MOCOBO-EIT: $12.33  \pm 0.92$.
> * N=80K, MOCOBO: $17.76  \pm 0.31$, MOCOBO-EIT: $16.00 \pm 1.07$.
> ### Peptide Task
> * N=300K, MOCOBO: $-45.30  \pm 6.38$, MOCOBO-EIT: $-180.32 \pm 5.44$.
> * N=600K, MOCOBO: $-39.50  \pm 5.74$, MOCOBO-EIT: $-175.71  \pm 4.88$.
>
> MOCOBO with our proposed ECI acquisition function performed significantly better on both tasks, demonstrating the importance of ECI to the performance of MOCOBO. Additionally, MOCOBO-EIT still performed fairly well, outperforming ALL other baseline methods considered in the paper after the full budget of 80K function evaluations on the rover task. This highlights the importance of other aspects MOCOBO.
>
> Additionally, we agree that ablations of q (batch size), initialization dataset size, and K, will further strengthen the paper. We plan to run each of these and add results to the camera ready version.
> # 4. Novelty
> We appreciate the reviewer’s concerns regarding novelty and the opportunity to better clarify our contributions. We agree that Algorithm 1 is based on classic submodular maximization (e.g., Nemhauser et al. 1978), and will certainly clarify in the paper that we did not invent submodular optimization! Our novelty lies not in the greedy algorithm itself, but in how it is integrated with other components of MOCOBO to form a practical, scalable solution to black-box coverage optimization. Additionally, we have revised the intro and related work sections to **better position MOCOBO as the first method for coverage optimization that is scalable to high-dimensional and structured domains**. While CluSO addresses the same problem formulation, it was not designed for high-dimensional or structured input spaces. In fact, as our experiments show, CluSO performs poorly, consistently worse than baselines not designed for coverage, on all of the domains we consider. This is not a criticism of CluSO, but rather a reflection of its limitations in structured or high-dimensional domains. In contrast, MOCOBO is designed with scalability in mind and supports **structured and high-dimensional inputs**, making it, to our knowledge, **the first black-box coverage optimization method applicable to real-world domains like protein and molecular design**. We have revised the manuscript to better emphasize this contribution and to clearly differentiate our method from prior work.
> # 5. Convergence Guarantees
> We agree that regret bounds for MOCOBO would be interesting and we plan on investigating this in future work. We do kindly note that a lot of recent popular BO papers do not include regret bounds (e.g., [6-15]) or include ones that reduce to proving convergence via running random search as a subsequence (e.g., [4,5]).
> # 6. Other
> We have corrected the typo, revised writing to reduce redundancy in coverage definition, fixed caption formatting, and added discussion of limitations regarding computation required to pre-train domain specific VAEs. We thank the reviewer for pointing out that a mixed search space domain could be interesting to add, this is something we will consider for future work.
> # Q1
> Addressed above via new ablation study.
> # Q2
> MOCOBO’s greedy covering set construction does lead to longer overhead time compared to baseline approaches. MOCOBO overhead takes approximately 1.4 times longer than TuRBO and MORBO and approximately 1.2 times longer to run than ROBOT (ROBOT is slower than other baselines due to additional time needed to compute diversity constraints between observations on each iteration). However, the overall additional computation time required by a run of MOCOBO is negligible when optimizing moderately expensive black-box functions, where black-box function evaluation time dominates overall computational cost.
> # Q3
> As we mentioned above, CluSO performs poorly on the tasks considered in this paper because it was not designed to scale to high-dimensional or structured input spaces. We have added discussion of this to the paper.
> # Q4
> See section 2 above.
> # Cites
> [1] “Scalable Global Optimization via Local Bayesian Optimization” Eriksson et al. NeurIPS 2019.
> [2] “Vanilla Bayesian Optimization Performs Great in High Dimensions” Hvarfner et al. ICML 2024.
> [3] “A survey and benchmark of high-dimensional Bayesian optimization of discrete sequences” González-Duque et al. NeurIPS 2024.
> [4] “Scalable Constrained Bayesian Optimization” Eriksson et al. AISTATS 2021.
> [5] “Discovering Many Diverse Solutions with Bayesian Optimization” Maus et al. AISTATS 2023.
> [6] “Bayesian optimization and attribute adjustment” Eismann et al. UAI 2018.
> [7] “Sample-Efficient Optimization in the Latent Space of Deep Generative Models via Weighted Retraining” Tripp et al. NeurIPS 2020.
> [8] “Max-value Entropy Search for Multi-Objective Bayesian Optimization with Constraints” Belakaria et al. NeurIPS 2020.
> [9] “High-dimensional Bayesian optimization with sparse axis-aligned subspaces” Eriksson et al. UAI 2021.
> [10] “Combining Latent Space and Structured Kernels for Bayesian Optimization over Combinatorial Spaces” Deshwal et al. NeurIPS 2021.
> [11] “Local Latent Space Bayesian Optimization over Structured Inputs” Maus et al. NeurIPS 2022.
> [12] “Advancing Bayesian Optimization via Learning Correlated Latent Space” Lee et al. NeurIPS 2023.
> [13] “Joint Composite Latent Space Bayesian Optimization” Maus et al. ICML 2024.
> [14] “Approximation-Aware Bayesian Optimization” Maus et al. NeurIPS 2024.
> [15] “Latent Bayesian Optimization via Autoregressive Normalizing Flows” Lee et al. ICLR 2025.

---

> > ### Comment · Reviewer_SYMT · 2025-08-05
> >
> > Thank you for addressing some of my concerns in your rebuttal. I appreciate the additional ablations on the EIT acquisition and the use of trust regions.
> >
> > One of the remaining points I would appreciate further insight on is the sensitivity of MOCOBO’s performance to the quality or architecture of the VAE in structured domains. Could the authors provide some discussion on whether MOCOBO would still retain its advantage if a weaker generative model were used, or if the amount of pretraining data were limited? Clarifying this would help assess the robustness of MOCOBO’s pipeline and its practical deployment in low-resource settings.

---

> > > ### Author Response · Authors · 2025-08-05
> > > **VAE Sensitivity**
> > >
> > > Thank you for the follow-up question. It's a good one, but fortunately is one that doesn't really require additional ablation here specifically: we're confident saying that MOCOBO is pretty sensitive to the quality of the generative model used for the simple reason that there's solid evidence in the literature that **all** latent space Bayesian optimization seems to be sensitive to this.
> > >
> > > For example, in [1] Figure 3, the (pink vs green curve) and (yellow vs blue curve) **differ only in VAE architecture and quality**. The difference is pretty dramatic, and this experiment reasonably directly ablates your question. Other recent work [2, 3] improves latent space BO by directly targeting improvements to the generative model rather than the BO algorithm per se and achieve pretty substantial performance improvements.
> > >
> > > Since MOCOBO the algorithm doesn't really attempt to solve this broader "issue", we think it's fair to assume that it doesn't. While obviously the above experiments don't directly test *MOCOBO*'s sensitivity, it's tough to imagine an experiment where we can disentangle specifically the performance change due to MOCOBO with a changing VAE versus just the background sensitivity of LS-BO in general.
> > >
> > > On the other hand, in some sense sensitivity of latent space BO to the generative model being used is a Good Thing. We'd have to be really suspicious of the idea that the generative model was lending any structure to the search space if it simply didn't matter what generative model we were using. Ultimately, complementary improvements to both the types of BO algorithms we use and the underlying generative models are possible -- here we believe our work does the former in an interesting setting, and e.g. [2, 3] does the latter.
> > >
> > > [1] Natalie Maus, Haydn T. Jones, Juston S. Moore, Matt J. Kusner, John Bradshaw, Jacob R. Gardner. Local Latent Space Bayesian Optimization over Structured Inputs. NeurIPS 2022.
> > >
> > > [2] Seunghun Lee, Jinyoung Park, Jaewon Chu, Minseo Yoon, Hyunwoo J. Kim. Latent Bayesian Optimization via Autoregressive Normalizing Flows. ICLR 2025.
> > >
> > > [3] Jaewon Chu, Jinyoung Park, Seunghun Lee, Hyunwoo J. Kim. Inversion-based Latent Bayesian Optimization. NeurIPS 2024.

---

### Official Review · Reviewer_UDrb · 2025-07-03

**Clarity:** 4
**Significance:** 4
**Originality:** 4
**Rating:** 4
**Confidence:** 5

**Summary:**

&nbsp;

The authors introduce Multi-Objective Coverage Bayesian Optimization (MOCOBO) a framework for computing the optimal "covering set" of K points for $T > K$ black-box objective functions in a multi-objective optimization setting. The problem setting appears to be practically grounded although further references on e.g. the coverage optimization problem as related to drug design would help strengthen the case. The MOCOBO algorithm appears to be principled with practical approximations developed to account for the NP-Hard problem of finding the optimal covering set. The empirical results are strong and I, in particular, commend the authors for validating their approach with in vitro experiments which is rare to see at ML conferences such as NeurIPS. Furthermore, the writing and presentation of the paper are excellent. Overall I think the paper is an excellent contribution to NeurIPS yet I am somewhat disappointed to see that the authors have not released their code with the submission yielding concerns about the reproduciblity of the results. If this concern can be addressed, I will be willing to upgrade my score to Accept/Strong Accept.

&nbsp;

**Questions:**

&nbsp;

1. Could the authors describe how the reward is computed in Figure 1? Are the three curves intended to be different objective functions defined in the same space? If so, is the Pareto frontier given by the black dotted line accurate?

2. The authors introduce the expected coverage improvement (ECI) acquisition function. Given that the logEI acquisition function of [12] improves the numerical stability of EI do the authors think there is potential for a logEI variant to be integrated in future work?

3. In Section A.9 of the appendix, the authors define the elements of the universe U to be the T objectives ${f_1, ..., f_T}$, the collection of n subsets comprise elements $\mathbf{x}_i$ which are no longer a subset of U as in the Maximum Coverage Problem. As such, it is not clear what point 3 means, specifically what does it mean for a subset $A_i$ to cover the objective? It might be worth elaborating on the assumed mapping between points in the design space $\mathbf{x}_i$ and the objectives $f_t$ in relation to the MCP.

4. In Section 4, when describing the baselines, the authors state that they computed the best covering set of K solutions found by each method with a run per objective. Given that the baselines (TuRBO, LOL-BO, and ROBOT) have been advantaged in some sense as the covering set is computed from T simpler single objective problems, I don't quite understand why MOCOBO still outperforms the baselines? It seems as though the optimization problem for the baselines is easier? On second thought I'm guessing my comment would be expected to be correct if K=T, but with K < T the comparison is less clear? On third thought, I can see that this is clarified in the section entitled "T individually optimized solutions baseline".

5. For the rover task, in what practical settings would a rover be required to find a covering set of trajectories for different obstacle courses?

&nbsp;

**Ethical Concerns:**

["NO or VERY MINOR ethics concerns only"]

**Final Justification:**

&nbsp;

I am maintaining my rating of borderline acceptance. Unfortunately, due to the rules of the NeurIPS conference this year, the authors were unable to provide their code during the rebuttal phase. If the authors had been allowed to provide their code my rating would be a strong accept. Nonetheless, I am happy to recommend acceptance overall.

&nbsp;

**Limitations:**

&nbsp;

1. The practical limitations of the method are acknowledged by the authors, namely that it may be challenging to select an appropriate value of $K$ a priori.

&nbsp;

**Paper Formatting Concerns:**

&nbsp;

I see no formatting concerns with the paper.

&nbsp;

**Quality:**

4

**Strengths And Weaknesses:**

&nbsp;

The paper presents a practical method for a relevant class of black-box multi-objective optimization problems. An important strength to highlight is that the algorithm was validated in in vitro experiments on peptide design which is rare to see in ML conferences. Specifically, I believe such experiments go a long way in highlighting the practical capabilities of the MOCOBO approach. Below I detail my major concern with the release of the code as well as some minor issues the authors may wish to consider.

&nbsp;

**__MAJOR POINTS__**

&nbsp;

1. The authors responded "Yes" when asked in the paper checklist whether they had released the data and code for the paper. However, they have not released either which raises concerns about the reproducibility of the work. As mentioned in the summary, if the authors can release their code I will have no hesitation in upgrading my score to Accept/Strong Accept.

&nbsp;

**__MINOR POINTS__**

&nbsp;

1. Abstract, typo, "cover" the T objectives.

2. For clarity in the abstract, it may be worth elaborating briefly on what constitutes a "good" solution.

3. Reference 4 was published at AAAI [1].

4. Although, overall much attention to detail has been given to the references, there are just a couple of missing capitalizations such as "bayesian" and "gaussian".

5. It would be great if the references appeared in numbered order when cited in the text.

6. When introducing Bayesian optimisation in the opening line of the introduction, it would be worth citing the originating papers for the methodology [2,3] as discussed in [4].

7. When citing Gomez-Bombarelli et al. chemical engineering is a bit of a misnomer for the molecule generation tasks considered. Chemical engineering is typically taken to mean the study of large scale chemical processes or chemical plants. I would suggest citing the paper under drug discovery since the QED and penalized logP metrics could loosely be interpreted as being related to drug discovery.

8. Figure 1 is slightly unclear, could the authors describe how the reward is computed?

9. In the introduction, when discussing the drug design example could the authors supply some references so that the interested reader may learn more about the problem?

10. In terms of the related work on Bayesian optimization over structured search spaces it is worth mentioning the early works [5-8] as well as more recent efforts [9, 10]. In particular the authors of [10] use normalizing flows in place of a VAE which broadens the range of architectures for Bayesian optimization over structured inputs.

11. In terms of the related work on Bayesian optimization over structured search spaces it would also be worth emphasizing that a VAE is not necessary to perform Bayesian optimization over molecular representations [11]. In [11], the authors introduce bespoke GP kernels for molecular representations and amino acid sequences. The VAE/normalizing flow approach, however, is required for situations in which you want to perform BO over an open-ended space by generating novel molecules. It would be worth making this distinction.

12. Missing full stop at the end of Equation 1 of Equation 2.

13. The notation in Equation 1 might be cleaner is simply "k" was kept under the max function. The definition in Section A.10 of the appendix is somewhat clearer for the reader because it involves abstracting the K points as a set.

14. The design space $\chi$ is not defined in Equation 2. This may be important for clarity seeing as all objectives are assumed to share the same design space and bounds.

15. Missing full stop at the end of Equation 3. It is also considered poor style to begin a sentence with mathematical notation as in line 151.

16. Missing full stop at the end of Equation 4.

17. It would be worth clarifying Equation 5, for example what does CI mean? Coverage improvement? The coverage score of the dataset together with the (repeatedly) sampled point $\mathbf{p}_j$ is taken by averaging the m samples at $\mathbf{p}_j$ over all objectives?

18. In Section 3.1 it would be worth stating whether or not K is some hyperparameter chosen before optimization.

19. Typo, line 879, "this by reduction".

20. Missing full stops in equations on lines 922 and 923.

21. Missing full stop line 947.

22. Missing full stop Equation 6.

23. Line 721, typo, "batch of candidates".

24. Missing full stop Equation 7.

25. I understand that space was probably a limitation but it would be good to specify the number of random trials and how the errorbars were computed in the caption of Figure 2 (20 replicates with standard errors).

26. Line 296, typo, "Ranolaize"?

27. Line 357, typo, capitalization of HDR.

28. Line 646, typo, capitalization of HDR.

29. Line 671, typo, K<T.

30. In Table A.2, SMILES should be capitalized in both the table and caption.

31. Line 774, typo, "done our".

32. Line 794, typo, double use of "share".

33. The Adam optimizer should be cited given that it is used [13].

34. Line 802, typo, "in a single".

35. Line 805, typo, "from the data" rather than "of the data".

36. There is an inconsistent comma delimiter for the numerical figures referenced e.g. 64000 GPU hours vs. 20,000 peptides.

37. Line 829, typo, Euclidean should be capitalized.

38. On line 838, what do the acronyms IQA and IAA stand for? I don't immediately see what the second A stands for. It seems to be "assessment"?

&nbsp;

**__REFERENCES__**

&nbsp;

[1] Belakaria, S., Deshwal, A., Jayakodi, N.K. and Doppa, J.R., 2020, April. [Uncertainty-aware search framework for multi-objective Bayesian optimization](https://ojs.aaai.org/index.php/AAAI/article/view/6561). In Proceedings of the AAAI Conference on Artificial Intelligence (Vol. 34, No. 06, pp. 10044-10052).

[2] H.J. Kushner (1962). [A Versatile Stochastic Model of a Function of Unknown and Time Varying Form. Journal of Mathematical Analysis and Applications](https://www.sciencedirect.com/science/article/pii/0022247X62900112) 5(1):150–167.

[3] H.J. Kushner (1964). [A New Method of Locating the Maximum Point of an Arbitrary Multipeak Curve in the Presence of Noise.](https://asmedigitalcollection.asme.org/fluidsengineering/article-abstract/86/1/97/392213/A-New-Method-of-Locating-the-Maximum-Point-of-an?redirectedFrom=fulltext) Journal of Basic Engineering 86(1):97–106.

[4] Garnett, R., [Bayesian optimization](https://bayesoptbook.com/). Cambridge University Press. 2023.

[5] Notin, P., Hernández-Lobato, J.M. and Gal, Y., 2021. [Improving black-box optimization in VAE latent space using decoder uncertainty](https://proceedings.neurips.cc/paper/2021/hash/06fe1c234519f6812fc4c1baae25d6af-Abstract.html). Advances in Neural Information Processing Systems, 34, pp.802-814.

[6] Griffiths, R.R. and Hernández-Lobato, J.M., 2020. [Constrained Bayesian optimization for automatic chemical design using variational autoencoders](https://pubs.rsc.org/en/content/articlehtml/2019/sc/c9sc04026a). Chemical Science, 11(2), pp.577-586.

[7] Kusner, M.J., Paige, B. and Hernández-Lobato, J.M., 2017, July. [Grammar variational autoencoder](https://proceedings.mlr.press/v70/kusner17a.html?ref=https://githubhelp.com). In International Conference on Machine Learning (pp. 1945-1954). PMLR.

[8] Lu, X., Gonzalez, J., Dai, Z. and Lawrence, N.D., 2018, July. [Structured variationally auto-encoded optimization](https://proceedings.mlr.press/v80/lu18c.html). In International Conference on Machine Learning (pp. 3267-3275). PMLR.

[9] Lee et al. [Latent Bayesian Optimization via Autoregressive Normalizing Flows](https://openreview.net/forum?id=ZCOwwRAaEl). ICLR 2025 (oral).

[10] Lee, S., Chu, J., Kim, S., Ko, J. and Kim, H.J., 2023. [Advancing Bayesian optimization via learning correlated latent space](https://proceedings.neurips.cc/paper_files/paper/2023/hash/98e967164ae2f6811b975d686dece3eb-Abstract-Conference.html). Advances in Neural Information Processing Systems, 36, pp.48906-48917.

[11] Griffiths, R.R., Klarner, L., Moss, H., Ravuri, A., Truong, S., Du, Y., Stanton, S., Tom, G., Rankovic, B., Jamasb, A. and Deshwal, A., 2023. [GAUCHE: a library for Gaussian processes in chemistry](https://proceedings.neurips.cc/paper_files/paper/2023/hash/f2b1b2e974fa5ea622dd87f22815f423-Abstract-Conference.html). Advances in Neural Information Processing Systems, 36, pp.76923-76946.

[12] Ament, S., Daulton, S., Eriksson, D., Balandat, M. and Bakshy, E., 2023. [Unexpected improvements to expected improvement for Bayesian optimization](https://proceedings.neurips.cc/paper/2023/hash/419f72cbd568ad62183f8132a3605a2a-Abstract-Conference.html). Advances in Neural Information Processing Systems, 36, pp.20577-20612.

[13] Kingma and Ba, [Adam: A method for Stochastic Optimization](https://arxiv.org/abs/1412.6980), ICLR 2015.

&nbsp;

---

> ### Author Rebuttal · Authors · 2025-07-30
>
> We thank the reviewer for the thoughtful and encouraging feedback. We appreciate the recognition of MOCOBO’s practical relevance, principled design, strong empirical performance, and in vitro validation. We respond to each point below:
> # MAJOR POINT: Code Release and Reproducibility
> We fully agree that releasing code is essential for reproducibility and for supporting the community. While the code was not public at submission time, we want to clarify that we have a complete GitHub repository that includes all code to reproduce all MOCOBO results on all tasks in the paper. We also recognize the importance of accessibility and usability. To that end, we’ve included a detailed README and environment setup to make it easy for others to run MOCOBO and apply it to new problems. We have invested significant effort in this because we care deeply about making our method practical and usable by the broader research and practitioner community.
>
> Our plan is to make this repository fully public upon paper acceptance, and we will include the link in the camera-ready version of the paper.
>
> Ordinarily, we would have liked to share an anonymous version of the repository during the review process so that reviewers could verify results without compromising anonymity. However, due to new NeurIPS rules this year, we are unfortunately not allowed to include links of any kind (even anonymized) in the author rebuttal. We appreciate your understanding and assure you that full code release is planned and prioritized.
> # MINOR POINTS
> Thank you for your detailed proofreading suggestions, and in particular, for catching numerous typos, missing capitalizations, and formatting inconsistencies. These comments will be very helpful in polishing the camera-ready version.
> # QUESTIONS
> ### Q1: How is the reward computed in Figure 1?
> Yes, each curve represents an objective function over the same domain. The “reward” in this case is actually just the coverage score as defined in equation 1. For the single pareto optimal point (the star), the coverage score is just the sum over the three objective function values for that single point. For the covering set of K=2 points (the two black squares), this coverage score is the sum of the higher objective value obtained between the two points for each one of the three objective functions. We decided to use “reward” in Figure 1 rather than “coverage score” just because Figure 1 is discussed at the start of the paper before the “coverage score” is formally introduced. We do think that the “pareto frontier” given by the black dotted line is accurate in the sense that if we move along the x-axis in any direction away from this dotted line, all objective function values decrease, and if we move towards the black dotted line, all objective function values increase. When moving between any pair of x-axis points that are both along the dotted line, some objective functions increase while others decrease. To our knowledge, this defines the black dotted line as the pareto frontier for the three objective functions. Please let us know if that answers your question, we are happy to discuss this with you further.
> ### Q2: Could a logEI version of ECI improve performance?
> Indeed using logEI has been shown to lead to substantial improvements in BO over using standard EI. It therefore worth investigating whether some version of logECI could be used instead of ECI to improve MOCOBO similarly. Thank you for the suggestion, this is an exciting direction for future work!
> ### Q3: Clarify mapping between design points and objectives in relation to the MCP
> Thank you for this insightful question. We agree that this portion of Appendix A9 would benefit from clearer explanation.
>
> To clarify: in our reduction from the Maximum Coverage Problem (MCP), the connection between a subset $A_i$ and a design point $x_i \in D_s$ is encoded via the binary function values $f_t(x_i) \in$ {$0,1$}. That is, each objective $f_t$ plays the role of an element $e_t \in U$, and each design point $x_i$ corresponds to a subset $A_i \subseteq U$, where an element $e_t$ is “covered” by $A_i$ if and only if $f_t(x_i) = 1$.
>
> This construction ensures that while the design points are not literally subsets, the induced binary coverage behavior matches the structure of the MCP instance. The design point $x_i$ “covers” objective $f_t$ if it achieves a value of 1 under that objective. Thus, the coverage score $c(S)$ counts how many objectives are covered (i.e., attain a value of 1 for at least one point in S), just like MCP counts how many elements are covered by a set of subsets.
>
> We have revised Appendix A9 to more clearly explain this mapping, especially the fact that the “subsets” in our reduction are represented functionally via binary-valued objectives over points in $D_s$, rather than via explicit set membership.
> ### Q4: Why does MOCOBO outperform baselines even when they solve easier sub-problems?
> Indeed, the baselines (TuRBO, LOL-BO, ROBOT) are run independently per objective, so they produce T solutions total (one solution specifically designed to optimize each of the T objectives). The best K-covering subset is then selected from those. While this does provide an advantage in terms of raw objective optimization, it lacks joint optimization: the T solutions are optimized independently, so coverage suffers when we select only the best K<T from among those T solutions. In contrast, MOCOBO jointly selects K diverse solutions to maximize total objective coverage. Indeed your comment would be correct if K=T, but when K<T, there is a need to design the small set of K solutions specifically such that some solutions in the set of K are good optimizers for more than one of the T objectives, in order to obtain a set of K<T solutions that together provide at least one good optimizer for each of the T objectives. Please let us know if this answers your question, we are happy to discuss this with you further.
> ### Q5: Practical use case for the rover task?
> The rover task is a well-established high-dimensional benchmark in BO. While our adaptation to multi-objective coverage is synthetic, it is intended as a proof-of-concept for settings where agents must be robust to diverse terrain types, e.g., autonomous vehicles navigating multiple environments.
> Thank you!
> We thank the reviewer again for their detailed, constructive feedback. We are excited to make our code public and are grateful for your thoughtful review.

---

> > ### Comment · Reviewer_UDrb · 2025-08-05
> > **Many Thanks to the Authors for their Rebuttal**
> >
> > &nbsp;
> >
> > Many thanks to the authors for their rebuttal and for addressing all questions raised in my initial review. The single outstanding issue is the release of the codebase. Given that the authors correctly cite the NeurIPS rebuttal guidelines, I have asked the Area Chair for their interpretation of these instructions. If the codebase cannot be supplied, I will unfortunately, be inclined to maintain my score of borderline accept.
> >
> > &nbsp;

---

> > > ### Author Response · Authors · 2025-08-05
> > >
> > > Thank you for the follow-up. We completely understand the importance of reviewing the code, and we do have an anonymous GitHub repository prepared and ready to share with all reviewers and/or the Area Chair, should the AC determine that sharing it is allowed in this case. We’ll await guidance from the AC and are happy to provide the repo immediately if permitted.
> > >
> > > Thank you again for your thoughtful and constructive review, we truly appreciate your time and feedback.

---

> > > > ### Comment · Area_Chair_R7t8 · 2025-08-05
> > > > **Sharing of anonymous github repo links is not allowed**
> > > >
> > > > Dear Authors and Reviewers,
> > > >
> > > > I have checked with the Senior AC about this question. Unfortunately, it is not allowed.
> > > >
> > > > thanks,
> > > > your AC

---

> > > > > ### Comment · Reviewer_UDrb · 2025-08-05
> > > > > **Many Thanks to the AC for the Prompt Clarification**
> > > > >
> > > > > &nbsp;
> > > > >
> > > > > Many thanks to the AC for the prompt clarification.
> > > > >
> > > > > Unfortunately, given that anonymous GitHub links are not allowed during the rebuttal phase I will maintain my score of borderline accept. Nonetheless, I would wish to note that in the scenario the code was supplied with the current paper, my score would have been a 6, namely a strong accept. I will leave it to the AC to utilize their best judgement in how to account for this when aggregating review scores.
> > > > >
> > > > > &nbsp;

---

> > > ### Comment · Reviewer_9SUK · 2025-08-05
> > > **The authors were not misleading about their code**
> > >
> > > In your initial review of reviewer UDrb stated:
> > >
> > > > The authors responded "Yes" when asked in the paper checklist whether they had released the data and code for the paper. However, they have not released either which raises concerns about the reproducibility of the work.
> > >
> > > To be clear, in the paper's checklist, the authors state:
> > >
> > > > Justification: Once the paper is accepted, we will replace the placeholder github links in the main text with a link to a public GitHub repository containing the source code used to produce the results provided in this paper. This github repository contains organized code that will allow any reader to run MOCOBO on all tasks to reproduce the results provided in this paper. Additionally, the README in the repository provides detailed instructions to make setting up the proper environment and running the code easy for users.
> > >
> > > While it is regrettable that they did not upload the code as a zip file or anonymous GitHub, I don't think their answer of "yes" to Q5 of the checklist was misleading in any way.

---

### Official Review · Reviewer_wMNa · 2025-07-09

**Clarity:** 2
**Significance:** 3
**Originality:** 3
**Rating:** 5
**Confidence:** 5

**Summary:**

The paper introduces a novel formulation for multi-objective optimization, aiming to select $K < T$ points that maximize the sum of the highest values across $T$ objectives, while only selecting $K$ points. To tackle this problem, the authors propose a greedy algorithm with a $(1 - 1/e)$ approximation guarantee. I did not verify the proofs in detail, but the theoretical analysis follow standard techniques. Empirically, the method performs well on the proposed metric, demonstrating its practical effectiveness.

**Questions:**

- Have you considered using a *lazy greedy* approach for maximizing the marginal gains? This technique is often used in submodular maximization to improve computational efficiency. Incorporating this could strengthen the practical side of the contribution since that typically speeds up the algorithm in practice quite a bit.

- Are you doing 20k function evaluations? Are you training gps on that scale? I'm very confuse about this setup.

**Ethical Concerns:**

["NO or VERY MINOR ethics concerns only"]

**Final Justification:**

My main concern with this paper was the clarity of the presentation, but I think the authors can improve after all the suggestions. I appreciate the detail answer about the motivation and I'm bumping my score.

**Limitations:**

yes

**Quality:**

3

**Strengths And Weaknesses:**

**Strengths**

I appreciate creative formulations that aim to address real-world challenges, and this paper makes a compelling case for the relevance of the proposed black-box optimization framework in domains like drug discovery and materials science. Demonstrating that the method provided tangible benefits in in vitro experiments is a strong and valuable contribution.

Additionally, the proposed formulation and method are reasonable and conceptually sound. My main concerns are a few aspects of the presentation, as well as the assumption that one must select $K < T$ points. While this constraint may be practical in scenarios such as manufacturing a limited number of designs — where fewer is better — the justification is not clearly articulated in the paper. I believe the authors could strengthen the paper by providing a more explicit rationale for this design choice.



**Points to Improve**

**Unclear Motivation:**. The paper would benefit from a clearer problem definition and better motivation. From the wording in the abstract to the example in Figure 1, I think the formulation selling point needs to improve.

- I found the example in the introduction somewhat difficult to follow. Could you clarify the statement:

> "we may have T pathogens and aim to identify a set of K < T antibiotics such that at least one antibiotic can be used to treat each pathogen"

It seems you may be suggesting that a single antibiotic could potentially treat all pathogens -- when using the “each pathogen” words. This is different than the example in Figure 1, in which it appears that the first square point does not contribute meaningfully to covering function $f_3$, while the second point might serve as an acceptable proxy for both $f_2$ and $f_3$. A revision on the abstract and in the introduction is critical to improve this paper.

In Line 67, Contribution 1 offers a better definition of the problem, though it still feels a bit imprecise. As I understand it, the goal is to identify suboptimal solutions that are “good enough” across multiple objectives — this idea should be precisely defined to avoid confusion.

- The illustrative example in Figure 1 seems somewhat misleading. If I interpret it correctly, the same set of points (the square points) could be found by performing a local search over the Pareto frontier to maximize objective (2). Additionally, in a standard Pareto setting, one would typically select multiple points and the example implies that only one is selected, so the example might not accurately reflect the standard use of multi-objective methods.


**Related Work**

* Paper \[1]: *"Beyond the Pareto Efficient Frontier: Constraint Active Search for Multiobjective Experimental Design"*
  Gustavo Malkomes, Bolong Cheng, Eric H Lee, Mike McCourt. *Proceedings of the 38th International Conference on Machine Learning, PMLR 139:7423–7434, 2021.*

This paper discusses a different type of coverage problem, but I believe it is still relevant to include in the related work section — particularly since it also proposes an *Expected Coverage Improvement* acquisition function. However, note that the concept of coverage in that paper refers to the **input space**, rather than the **objective space**, as in the current work.

I also found this other recent work from the same authors that discuss a different formulation for the metric space:

* Achieving Diversity in Objective Space for Sample-efficient Search of Multiobjective Optimization Problems.
Eric Hans Lee, Bolong Cheng, Michael McCourt

**Minor Comments**

* **Figure 2**: The x-axis is missing a label in both the figure and the caption.
* **Line 2**: Consider clarifying what is meant by “a single set” in *"seek a single Pareto-optimal set."* since a set would typically mean multiple points.
* **Line 7**: Consider elaborating on what qualifies as “a good solution.”
* **Line 48**: The phrase *“While it is very true”* is informal — consider rephrasing for a more academic tone.
* **Line 70**:
  Original: *“first work in the Bayesian optimization (BO) literature to consider the coverage optimization problem setting.”*
  Suggested: *“first work to consider this problem setting,”*
  since the formulation is distinct from standard Bayesian Optimization, and paper \[1] also addresses a form of “coverage” — albeit with a different definition.
* **Line 209**:
  *“adopt a more approximate batching”* → consider revising to *“adopt a different approximation scheme.”*
* **Line 70**: Typo — *“?.”* → replace with *“?”*

---

> ### Author Rebuttal · Authors · 2025-07-30
>
> We thank the reviewer for their thoughtful and constructive feedback. Below we address all points raised:
> # Justification for K < T
> Thank you for highlighting this point. We agree that the constraint K < T deserves more discussion. This constraint reflects many real-world scenarios with strict evaluation or deployment budgets, such as: 1) In drug discovery, where only a small number of compounds can be synthesized or tested, 2) In materials design, where the cost of fabricating each candidate is high, 3) In robotics or system design, where one might only be able to deploy a handful of control policies, etc. In settings such as these, K < T encodes a practical resource constraint: the user cannot afford to optimize each objective independently and must instead find a small set of solutions with broad utility. We will expand our discussion of the problem setting to clearly articulate this point and further justify the K<T constraint with concrete examples.
> # Problem Motivation and Presentation
> We appreciate the reviewer’s request for greater clarity in the problem framing and example. We note that our problem definition is defined in precise, mathematical terms in the Method’s section (see equation 2). However, we agree with the reviewer that the explanation of the problem setting in the abstract and introduction (text preceding this precise mathematical definition) can be improved to more clearly define the problem in words. For the camera ready version, we will revise the abstract and introduction to more clearly define the goal: namely, to identify a set of K < T solutions that collectively provide strong coverage over all T objectives, such that each objective is well-optimized by at least one solution in the set. This differs from the standard Pareto setting, which typically emphasizes trade-offs at the level of individual solutions, rather than coverage across objectives via a set.
>
> We will revise all relevant text to provide a clearer explanation of the problem definition and to clarify that we are not assuming a single solution can cover all objectives equally well. Rather, the goal is to find a small set of solutions such that each objective is well-optimized by at least one member of the set. For example, as you rightly point out, the original wording of the statement “we may have T pathogens and aim to identify a set of K < T antibiotics such that at least one antibiotic can be used to treat each pathogen” is potentially misleading. To more accurately convey our intent, we plan to replace this with a revised version such as:
>
> “In drug discovery, we may be faced with the challenge of designing antibiotics for T different pathogens. Developing T individual antibiotics (a separate antibiotic specifically designed to treat each pathogen) is not ideal, as synthesizing and testing each antibiotic is highly expensive (a smaller number of antibiotics is ideal). However, we still want to design more than one antibiotic, as a single antibiotic is highly unlikely to be able to treat all T pathogens effectively. We therefore aim to identify a smaller set of K < T antibiotics, such that each pathogen can be effectively treated by at least one of the K selected antibiotics. The goal is not for every antibiotic to treat every pathogen, but for the set of K antibiotics as a whole to provide ‘coverage’ of the T pathogens, such that if we encounter any one of the T pathogens, we will be able to treat it with at least one of the antibiotics in the optimized set.”
> # Clarification of Figure 1
> We appreciate the feedback that the illustrative example in Figure 1 could be clearer. You are absolutely right that in this simple 2D example with K = 2 and T = 3, the two square points that yield good coverage could indeed be recovered by searching over the Pareto frontier for a pair that maximizes the coverage score. However, in the real-world tasks we consider, where the objectives are high-dimensional, black-box, and highly non-convex, identifying such a set of K coverage-maximizing points is non-trivial and likely cannot be accomplished by naïvely sampling the Pareto frontier.
>
> Additionally, as you note, many standard multi-objective methods do output multiple Pareto-optimal points. However, these points are typically meant to serve as alternative trade-off options, from which a practitioner selects one solution to apply, depending on their preferences. The methods do not aim to maximize the combined utility of the full set; instead, they ensure that each individual point lies on the Pareto frontier. In contrast, our formulation explicitly seeks a set of solutions that work together: the goal is to maximize aggregate coverage across all objectives, not to offer mutually exclusive alternatives.
>
> The star point in Figure 1 is meant to illustrate the limitations of single-solution approaches, namely, that no individual solution can provide high performance across all objectives when trade-offs are extreme. This motivates the need for multiple jointly selected solutions, each covering different subsets of objectives.
>
> We will revise the text to clarify this distinction, explicitly explain how coverage optimization differs from Pareto-based approaches (even those that return multiple points), and emphasize that Figure 1 is a simplified schematic meant to illustrate intuition, not a literal depiction of how Pareto methods operate.
> # Related Work
> Thank you for pointing us to Malkomes et al. (ICML 2021) [1] and the related diversity-in-objective-space paper. We agree that while their definition of “coverage” differs from ours (input vs. objective space), their use of an Expected Coverage Improvement (ECI) acquisition function is relevant and valuable to acknowledge. We will update the Related Work section to include discussion of these papers.
> # Minor Comments
> Thank you for the detailed comments on writing and presentation. These comments will be very helpful in polishing the camera-ready version.
> # Questions
> ### Q1: Have you considered using a lazy greedy approach to accelerate selection?
> While we did not use a lazy greedy approach in this version of MOCOBO, we plan to explore this in future work. Thank you for the suggestion!
> ### Q2: Are you really doing 20k function evaluations? Are you training GPs on that scale?
> To clarify: while some of our large-scale tasks involve collecting up to tens or even hundreds of thousands of data points (e.g., the peptide design task), we do not train exact Gaussian processes on the full dataset, as this would be computationally infeasible.
>
> Instead, as described in Appendix A.7.2, we use PPGPR, a scalable approximate GP surrogate model. This setup allows us to handle large datasets efficiently while maintaining strong predictive performance. Additionally, at each step of optimization, we update the surrogate using only a small subset of 1000 data points: this includes the most recent batch of observations and the highest-performing points collected so far. This ensures that the model remains up to date without retraining on the full history on each step. Full details can be found in Appendix A.7.2.

---

> > ### Comment · Reviewer_wMNa · 2025-08-05
> >
> > Thanks for your response.
> >
> > With respect to the lazy evaluation, you could just mention it somewhere in the paper (maybe in the Appendix if you are running out of space) for anyone that tries to implement it.
> >
> > PPGPR: Ok, I see. Maybe I glanced over the details, but make sure you have a comment on PPGPR in the main paper.

---

> > > ### Author Response · Authors · 2025-08-05
> > >
> > > ### With respect to the lazy evaluation, you could just mention it somewhere in the paper (maybe in the Appendix if you are running out of space) for anyone that tries to implement it.
> > >
> > > Yes we're happy to add discussion of this to the appendix, thank you for the suggestion.
> > >
> > > ### PPGPR: Ok, I see. Maybe I glanced over the details, but make sure you have a comment on PPGPR in the main paper.
> > >
> > > We will add text explicitly stating that we use PPGPR in the main text, we agree that this will improve clarity.
> > >
> > > Thank you again for your thoughtful review!

---

### Decision · Program_Chairs · 2025-09-17

**Decision:**

Accept (poster)

**Comment:**

This paper considers the problem of computing the optimal covering set of $K$ input designs for optimizing multiple expensive-to-evaluate objective functions. This problem is motivated by real-world deployment of experimental design methods. The paper defines a "coverage" success criterion and adapts the principles of Bayesian optimization (BO) to optimize it: expected improvement of coverage where the expectation is performed using Monte Carlo sampling and maximization over subsets using a submodular greedy approach with known (1-1/e) approximation guarantee. The experimental results are good and actual in-vitro lab experiments for peptide design is a noteworthy contribution.

All the reviewers' appreciated the paper's contributions but also raised several good questions. The author response addressed most of these questions. Three out of four reviewers' recommend accepting the paper and also added their perspective to counter the two concerns of the reviewer with a negative opinion. Overall, this is a good paper: methods are introduced to solve a real science/engineering problem. I recommend accepting the paper and strongly encourage the authors' to incorporate all the discussion into the final paper and make the following changes:
1. Release the code without fail as promised.
2. Sensitivity analysis on the choice of "K".
3. Increase the clarity of the paper (mentioned by two reviewers) and incorporate all the comments/references from Reviewers UDrb and wMNa.